



# An annually resolved chronology for the Mount Brown South ice cores, East Antarctica.

Tessa R. Vance[1], Nerilie J. Abram[2], Alison S. Criscitiello[3], Camilla K. Crockart[4,1], Aylin DeCampo[5], Vincent Favier[6], Vasileios Gkinis[5], Margaret Harlan[4,1,5], Sarah L. Jackson[2], Helle A. Kjær[5], Chelsea A. Long[4], Meredith Nation[7,1], Chris T. Plummer[7,1], Delia Segato[8,9], Andrea Spolaor[8,9], and Paul T. Vallelonga[5]

[1]Australian Antarctic Program Partnership, Institute for Marine & Antarctic Studies, University of Tasmania, Battery Point 7004, Australia
[2]Research School of Earth Sciences, ARC Centre of Excellence for Climate Extremes and Australian Centre for Excellence in Antarctic Science, Australian National University, Canberra ACT 2601, Australia
[3]Department of Earth and Atmospheric Sciences, University of Alberta, Edmonton, Canada
[4]Institute for Marine & Antarctic Studies, University of Tasmania, Battery Point 7004, Australia
[5]Physics for Ice, Climate and Earth, Niels Bohr Institute, University of Copenhagen, Copenhagen, Denmark
[6]Université Grenoble Alpes, CNRS, IRD, Grenoble INP, IGE, 38000 Grenoble, France
[7]Australian Antarctic Division, Department of Climate Change, Energy, the Environment and Water, Kingston, Tasmania, Australia
[8]Institute of Polar Sciences, CNR-ISP, Campus Scientifico Via Torino 155, 30172 Mestre, Venice, Italy
[9]Department of Environmental Sciences, Informatics and Statistics, University Ca' Foscari of Venice, via Torino, 155 - 30172 Venice-Mestre, Italy

**Correspondence:** Tessa R. Vance (tessa.vance@utas.edu.au)

**Abstract.** Climate reconstructions of the last millennium rely on networks of high resolution and well-dated proxy records. This study presents age-at-depth data and preliminary results from the new Mount Brown South ice cores, collected at an elevation of 2,084 metres on the boundary of Princess Elizabeth and Kaiser Wilhelm II Land in East Antarctica. We show an initial analysis of the site meteorology, mean annual chemical species concentrations, and seasonal cycles including analysis

5  of a seasonal cycle in fluoride concentrations with a potential link to sea ice formation. The annually resolved chronologies were developed from this data using a site-specific layer-counting methodology which employed seasonally varying trace chemical species and water isotope ratios, combined with a volcanic horizon alignment approach. The chronologies developed include the 'Main' 295 m record spanning 1,137 years (873-2009 CE), and three surface cores spanning the most recent 39-52 years up to the surface age at the time of drilling (austral summer 2017/2018). Mean annual trace chemical concentrations are

10  compared to the Law Dome ice core further to the east and discussed in terms of atmospheric transport, and the uncertainty in the determination of annual horizons via layer counting is quantified. The MBS chronologies presented here - named MBS$_{2023}$ - will underpin the development of new palaeoclimate records spanning the past millennium from this under-represented region of East Antarctica.





## 1 Introduction

The variability of mid- to high-latitude climate over recent millennia is poorly understood in the Southern Hemisphere (Jones et al., 2016). In the Indo-Pacific sector of the Southern Ocean, data sparsity is particularly acute prior to the satellite era in 1979 due to the lack of inhabited land masses and meteorological stations. In addition, there are few millennia-scale, high resolution (seasonally to annually resolved) palaeoclimate records from ice core drilling efforts in the Indo-Pacific sector of East Antarctica (Jones et al., 2016; Stenni et al., 2017; Thomas et al., 2017, 2023). As an example, for the 100 degrees of longitude spanning 50-150° E there is only one existing millennial length coastal ice core record (Law Dome). This study introduces the Mount Brown South ice core, which will add a new long, coastal ice core to this region, and contribute to global efforts to expand the network of annually resolved Antarctic ice core records spanning the last 1-2 millennia (see https://pastglobalchanges.org/science/end-aff/ipics/white-papers).

This study presents the ice core age-at-depth scales (chronologies) derived from layer counting of seasonally varying chemical and isotopic species for the Mount Brown South (MBS) 'Main' ice core, an intermediate length record drilled over austral summer 2017/2018, as well as the three 20-25 m surface cores from the same drilling site and season. The MBS site was chosen after a comprehensive site selection effort that incorporated remote sensing and radar surveys across a number of promising ice core sites in East Antarctica (Vance et al., 2016). Site selection also incorporated data from a network of 30 short (5-15 metre) surface cores collected across coastal East Antarctica over the last three decades by the Australian Antarctic Program, including in the Mount Brown region (Smith et al., 2002; Foster et al., 2006). Seven desirable physical criteria were defined to locate multiple prospective East Antarctic sites.

1. Ice of up to 2000 years age at 300 m depth.

2. No or minimal likelihood of summer melt (to aid preservation of climate signals).

3. Mean annual snowfall accumulation of >0.25 metres ice equivalent (to enable sub-annual sample resolution).

4. Minimal local surface re-working (to maximise preservation of climate signals).

5. Site location on a ridgeline or dome to reduce ice advection through time.

6. An atmospheric link to the mid-latitude circulation of the Indian Ocean.

7. A record complementary (i.e. adds new information) to the existing coastal East Antarctic ice core array.

The Mount Brown South site was identified as one of four promising regions across the 50-120° E sector of coastal East Antarctica which falls within the Australian Antarctic Program's operations region. The chronologies presented here follow two initial studies: One of sea salt and snowfall accumulation rates over the 1975-2017 period, which combined data from the upper portion of the intermediate length record and the three 20-25 m surface cores drilled at the Mount Brown South site (Crockart et al., 2021). The second analysed seasonal and event-scale variability in water stable isotope ratios over the satellite era (Jackson et al., 2022). These prior studies show that for the satellite era, the new MBS ice core fulfills the stipulations in the



site selection study, albeit with a tendency toward higher annual accumulation rates than originally estimated in Vance et al. (2016).

## 1.1 Mount Brown South drilling site characteristics

The Mount Brown South site is located on the boundary of Princess Elizabeth and Wilhelm II Land, East Antarctica at 69.111°
S, 86.312° E and 2,084 m above sea level (Figure 1). Ice thickness in the region around the site has a mean value of approximately 2000 metres, while selecting for minimal elevation change and ice advection over the millennial timescales of interest was also taken into account during site selection (e.g. see Figure 3 in Vance et al. (2016)). Mean annual surface air temperatures (T2) for the MBS site derived from the Modele Atmospherique Region (MAR) (Agosta et al., 2019) are -27.9°C, while mean summer (DJF) surface air temperatures are -18.4°C. The nunatak of Mount Brown is located 62 km north of the drill site, while
the nearest permanently manned Antarctic station (Davis station) is 380 km to the west (Figure 1). The ice core site is 12.6 km WSW of a 10 m surface core ('MBS99') drilled in December 1998 (Smith et al., 2002; Foster et al., 2006). The cores described in this study are named 'MBS1718' to differentiate them from past/future ice drilling at this site. For simplicity in this study we will hereafter refer to the 2017/2018 ice cores as 'MBS' unless comparing to the earlier drilling effort.

    Crockart et al. (2021) and Jackson et al. (2022) found the snowfall accumulation regime at MBS to follow a seasonal cycle
of higher precipitation during the polar winter (March-October), and lower precipitation during December/January. Variability between accumulation records across the site was present, due to the prevailing easterlies and resulting surface features (Figure 1), which in some cases were found to be of equivalent height to the satellite era mean annual accumulation rate of 0.3 metres ice equivalent (m.i.e.) (Figure 2 and Crockart et al. (2021)). Coastal East Antarctica is subject to maritime moisture intrusions which result in intense precipitation events occurring over hours to days, a subset of which reach thresholds to be classified
as atmospheric rivers (Gorodetskaya et al., 2014; Turner et al., 2019; Wille et al., 2021). At MBS, intense precipitation events on average account for 44% of annual accumulation (Jackson et al., 2022). These events are strongly related to mid-latitude blocking in the southern Indian Ocean to the northeast of MBS, which channels warm and moist maritime air masses to the site (Jackson et al., 2022; Pohl et al., 2021; Udy et al., 2021, 2022).

    We derived estimates of site climatology from the regional climate model MAR (Modele Atmospherique Region) for the
nearest pixel to the MBS drilling site (86.4785°E, 69.2429°S, 2138 m asl). Only limited variability in prevailing wind speeds and directions occurs during each season. Slower wind speeds occur in DJF, which shows daily wind speeds only rarely exceed 12 m s$^{-1}$ (Figure 2). Higher precipitation is associated with winds directly from the east, while lower snowfall occurs when there is a southerly element to the easterly flow. At the time of drilling the MBS site showed numerous large surface features which were 10's of metres wide and hundreds of metres long aligned to approximately 80° (Figure 1 and Crockart
et al. (2021)). Other smaller features of a few to 10's of metres crossed these larger surface features and were aligned to around 110°, suggesting wind from these directions had occurred in the weeks to months prior to drilling. Given the seasonal cycle of accumulation at MBS displays higher accumulation during the polar winter (March/April to November), this suggests a relationship between accumulation and wind speed. For a deeper analysis of variability in isotopic ratios, temperature and

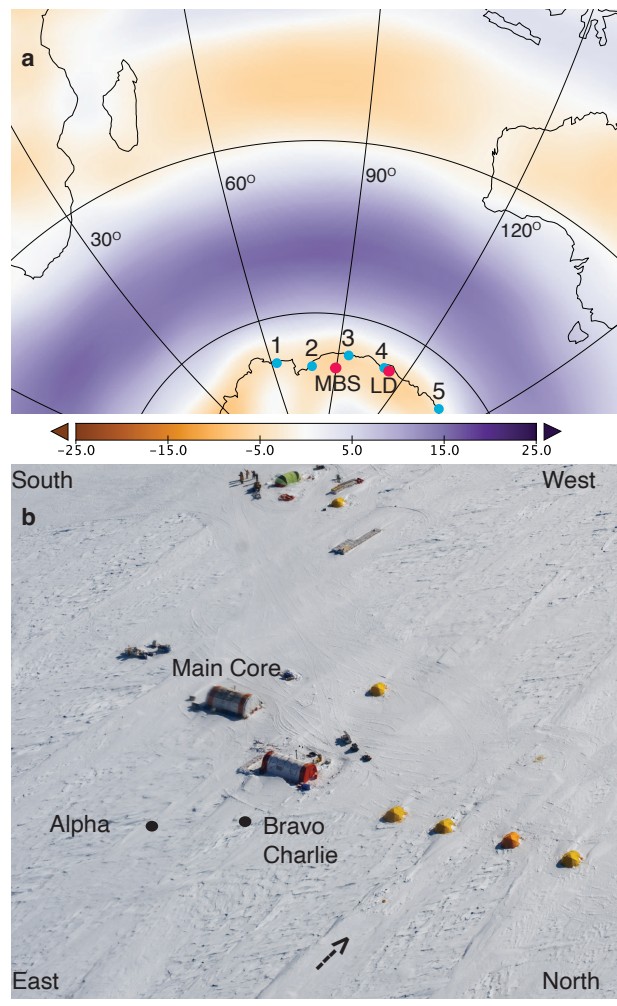

**Figure 1.** (a) The southern Indian Ocean and East Antarctic coast with the Mount Brown South ice core drill site shown (fuschia dot, MBS). Indian Ocean sector mean zonal wind over the satellite era (1979-2016) indicates the prevailing mid-latitude westerly wind stream (purple shading) and the polar easterlies around the coastline of Antarctica (orange shading) in m s$^{-1}$. The relative positions of the East Antarctic research stations Mawson (1), Davis (2), Mirny (3), Casey (4) and Dumont D'Urville (5) are indicated from west to east respectively (cyan dots). The Law Dome drill site is approximately 100 km east of Casey (fuschia dot, LD). (b) Aerial photo of the Mount Brown South site and drilling camp showing the relative locations of the Main core (drill tent) and surface cores Alpha and Bravo/Charlie, located 94 and 82 metres from the Main core/drill tent respectively. Large wind-blown surface features aligned to approximately 80° can be seen across the site, with smaller features aligned to approximately 110°, reflecting the predominant easterly airflow (note arrow at bottom of image showing predominant wind direction). See Crockart et al. (2021) and Jackson et al. (2022) for alternative site maps. Photo credit: Doug Westersund/Kenn Borek Air.



wind direction, see Jackson et al. (2022). No particular differences in wind speed and direction were observed between different
reanalysis products (e.g. MAR versus ERA 5) or from one decade to the next.

## 2  Methods

### 2.1  Drilling and Field processing

The MBS Main core and the three surface cores, Alpha, Bravo and Charlie, were drilled over three weeks in austral summer
2017/2018 (Table 1). A 1.87 m deep snowpit was also excavated, located within 5 metres of the Bravo and Charlie surface
cores. Snowpit temperatures were taken every 5 cm and ranged between -15°C and -24.7°C.

MBS Main was drilled using the Hans Tausen drill (Johnsen et al., 2007; Sheldon et al., 2014a) from 4.25 m below the
surface to the end of drilling at 294.785 m. At 93.6 m depth, wet drilling commenced with the use of ESTISOL 140 as drilling
fluid (Sheldon et al., 2014b; Talalay et al., 2014). The three shallow cores were drilled using a Kovacs Mark II ice core drill
coupled with a battery-powered electric motor. After drilling, each core sample was cut to 1 m lengths using a handsaw and
the length, diameter, and weight recorded for density calculations before storing in sealed polyethylene (LDPE) bags. Core
samples were stored in insulated boxes beneath the snow surface to keep frozen until being flown to freezer storage at Davis
Station, then transported by ship to Australia.

**Table 1.** Description and locations of the four Mount Brown South ice cores. MBS Alpha, Bravo and Charlie were drilled using the Kovacs
surface core system. The MBS Main core was drilled using a Hans Tausen Intermediate length system.

| Core ID | Position | Relative site location | Drilling date | Depth range (m) |
|---|---|---|---|---|
| MBS Main | 69.111° S, 86.312° E | Drill tent | 23 December 2017-15 January 2018 | 4.25-294.785 |
| MBS Alpha | 69.111° S, 86.315° E | 94 metres ENE of Main | 20 December 2017 | surface to 20.41 |
| MBS Bravo | 69.110° S, 86.314° E | 82 metres NE of Main | 14-15 January 2018 | surface to 20.225 |
| MBS Charlie | 69.110° S, 86.314° E | 82 metres NE of Main | 15-17 January 2018 | surface to 25.86 |

### 2.2  Core processing and sample preparation

Core sectioning and discrete sample preparation occurred in the -18° C ice core processing freezer laboratories at the Institute
for Marine & Antarctic Studies (IMAS) in Hobart. Here, detailed records and drawings including dimensions, breaks, drilling
and transport damage, wind crusts, and other stratigraphic features were made. The cores were then sectioned using a cleaned
bandsaw installed in our ice core freezer laboratory according to a planned cutting guide (Figure 3). Two 34 x 34 mm ice sticks
were taken along the length of the Main core for trace ion analysis; one for discrete trace ion chromatography and one for
Continuous Flow Analysis (CFA). Both CFA and discrete chemistry sticks were sectioned from the Charlie core as well, and
a discrete chemistry stick and isotopes strip only from the Alpha core. A surface strip for isotope analysis was sectioned from
the Bravo core, with the remainder preserved for persistent organic pollutant studies. Prior analysis of the Alpha and Bravo



**Figure 2.** Seasonal wind speed (m s$^{-1}$), frequency (as a percentage of time) and direction at the MBS site during high (>1 mm water equivalent d$^{-1}$) and low (<1 mm water equivalent d$^{-1}$) snowfall regimes from the Modele Atmospherique Region over the satellite era.



cores (which were drilled in close proximity) has shown they have very similar isotope records (Crockart et al., 2021) meaning the age-by-depth scale developed for the Alpha core can be easily mapped to the persistent organic pollutant records that will be developed from the Bravo core.

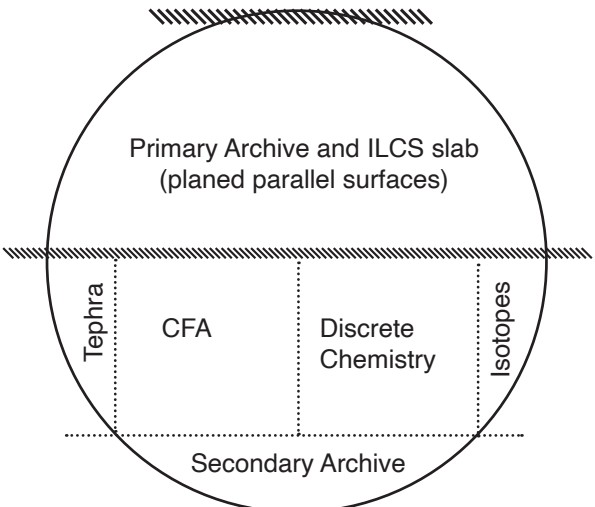

**Figure 3.** Sampling schematic for the primary analyses of the Mount Brown South Main ice core. Dotted lines indicate processing via bandsaw, while dashed lines indicate planed surfaces of the remaining primary archive prior to imaging via Intermediate Layer Ice Core Scanner (ILCS). CFA indicates the analysis stick for continuous flow analysis at the University of Copenhagen. The remaining primary and secondary archive sections have been stored under clean and low temperature conditions for future analyses.

Discrete trace chemistry samples at 3 cm resolution were produced under laminar flow in an ice core freezer using the clean procedures outlined in Plummer et al. (2012); Sanz Rodriguez et al. (2019); Crockart et al. (2021). For a standard bag length of 1 m, the final discrete sample was subsequently 4 cm (isotopes) or 3.5 cm (discrete chemistry), yielding 33 samples per bag (metre) of ice. The slightly shorter discrete chemistry final sample length resulted from the sample cleaning technique, which utilised a vice in a laminar flow hood to restrain the chemistry stick during sample preparation which resulted in a small
remaining portion, typically 0.5 cm, in the vice. Given the accumulation rates calculated for the satellite era in Crockart et al. (2021), this sampling resolution yields a mean discrete sample resolution of 10 samples $y^{-1}$. We also used proven sample melt/refreeze procedures on the discrete chemistry samples to preserve MSA concentrations in sample meltwater prior to analysis (Curran and Palmer, 2001; Abram et al., 2008; Roberts et al., 2009).

    Discrete water isotope samples were also sectioned at a resolution of 3 cm, except between $17 - 35$ m of the Main core
where they were taken at 1.5 cm resolution in order to investigate optimal sampling resolution for identifying annual layer horizons. Interlaboratory comparison and calibrations (Jackson et al., 2022) identified sufficient temporal resolution for annual layer counting was obtained by 3 cm sectioning.



## 2.3 Trace chemistry and water stable isotope analysis from discrete samples

Trace analysis of chloride ($Cl^-$), nitrate ($NO_3^-$), sulfate ($SO_4^{2-}$), methanesulfonic acid ($MSA^-$), fluoride ($F^-$), sodium
($Na^+$), magnesium ($Mg^{2+}$), calcium ($Ca^{2+}$) and potassium ($K^+$) were made on the 3 cm discrete samples using suppressed
ion chromatography. For detailed methods of discrete analysis see Jong et al. (2022) and references therein. In brief, we used
an Ionpac CS19 (Thermo Scientific™/Dionex™) cation analytical column to improve detection and separation of magnesium
and calcium peaks (Plummer et al., 2012), and an Ionpac AS15 anion column which greatly improved separation of fluoride
and MSA over previous methods and columns (Sanz Rodriguez et al., 2019). During analysis we noticed that the MSA data
between 20-93 m of the Main core exhibited unknown contamination affecting peak detection. This organic contamination
only affected separation of MSA. An additional 34 x 32 mm ice stick for this section was prepared for reanalysis, including 5
m of overlap (i.e 15 m – 93 m) to ensure sample reproducibility. An additional time series of interest for dating and alignment
with volcanic horizons - the sulfate to chloride ratio - was derived from the raw trace chemistry data. This ratio produces a
clear annual boundary at the MBS site, similar to Law Dome (Plummer et al., 2012; Jong et al., 2022; Crockart et al., 2021).

Identification of known pre-historical volcanic events is crucial to the analysis of error in layer-counted ice core records.
Given the three primary sources of $SO_4^{2-}$ to the Antarctic ice sheet (sea salt aerosol, biogenic activity and volcanic eruptions),
we identified volcanic events by estimating the non sea salt component of $SO_4^{2-}$ ($nssSO_4^{2-}$) according to Plummer et al.
(2012). We calculated a relative estimate of $nssSO_4^{2-}$ (equation 1) using the seawater ratio ($R$) of total $SO_4^{2-}$ to $Na^+$
concentrations to remove the bulk sea salt $SO_4^{2-}$ component. Note that our analytical outputs are in µEq $L^{-1}$, thus $R = 0.1201$.


$$[nssSO_4^{2-}] = [SO_4^{2-}{}_{total}] - R[Na^+{}_{total}] \tag{1}$$

Importantly, this relative estimate does not quantitatively calculate $SO_4^{2-}$ of volcanic origin at MBS. To do this, we would need
to remove the seasonal cycle (in order to estimate and remove the biogenic component) and also account for any local/regional
fractionation of the sea salt signal (for example, due to $SO_4^{2-}$ depletion from the formation of frost flowers on sea ice). This
fractionation would vary the value of $R$ slightly (e.g. see Plummer et al. (2012) and Palmer et al. (2002) for this fractionation
estimate at Law Dome) but is unnecessary for the identification of volcanic peaks in this study.

We undertook sequential inter-laboratory comparisons of discrete and CFA derived isotopes at the three isotope-capable
laboratories involved in this study - AAPP/IMAS in Hobart, Australia, the Australian National University in Canberra, Australia
and the Centre for the Physics of Ice, Climate, and Earth at the Niels Bohr Institute (University of Copenhagen) in Copenhagen,
Denmark. For the upper 73 metres of the Main core and all surface cores, the water isotope ratios of $\delta^{18}O$ and $\delta D$ were analysed
as discrete samples by cavity ring-down spectroscopy on a Picarro L2130-i water isotope analyser in Hobart. Then, discrete
Main core samples from the top to 110 m depth and the Charlie surface core (which included those samples already analysed
in Hobart) were shipped to Canberra and were analysed for $\delta^{18}O$, $\delta^{17}O$ and $\delta D$ on a Picarro L2140-i water isotope analyser.
These samples provide a large analytical inter-laboratory comparison between the Hobart and Canberra laboratories, as well as
a 16 metre interlaboratory comparison with CFA isotope data analysed in Copenhagen. See Table 2 for details of the isotope
analyses undertaken by depth and laboratory. Jackson et al. (2022) report excellent interlaboratory agreement (r>0.99 for all





parameters) suggesting negligible fractionation between melt and refreeze episodes of samples or interlaboratory differences between the various analysis campaigns. A manuscript detailing the full isotope dataset is under preparation.

**Table 2.** Details of the isotope sampling and analyses over the three laboratories involved in this study, including the overlapping sections allowing interlaboratory and analytical comparisons.

| Core ID | Depth range (m) | Analysis system | Laboratory | Analysis dates |
|---|---|---|---|---|
| MBS Alpha | surface-20.41 | Discrete 3cm | Hobart | 2018-2019 |
| MBS Bravo | surface-20.225 | Discrete 3cm | Hobart | 2018-2019 |
| MBS Charlie | surface-25.86 | Discrete 3cm | Hobart | 2018-2019 |
| MBS Charlie | surface-25.86 | Discrete 3cm | Canberra | 2019-2020 |
| MBS Main | 4.25-73.0 | Discrete 1.5-3cm | Hobart | 2018-2019 |
| MBS Main | 4.25-110.0 | Discrete 3cm | Canberra | 2019-2020 |
| MBS Main | 93.6-295.785 | CFA | Copenhagen | 2019 |

## 2.4 Continuous flow analysis of trace chemistry, peroxide, dust and water stable isotopes

Sample sticks (34 x 34 mm) for continuous flow analysis (CFA) sectioned from the MBS Main core and the Charlie surface core were shipped frozen to Copenhagen to be analysed via CFA at the Niels Bohr Institute (Kaufmann et al., 2008; Bigler et al., 2011). We used a modified version of the Copenhagen CFA system (Bigler et al., 2011; Kjær et al., 2022) during two campaigns, one which analysed the dry-drilled sections of the Main core (4-94m) and the Charlie core (CFA1: November 2018), and a second which analysed the deeper wet drilled section (95-295m) of the Main core (CFA2: November 2019).

Prior to melting, sample stick ends and breaks were cleaned by removing around 1 mm of ice using a microtome blade. The cleaned ice was then placed vertically on the melt head inside a small freezer. During melting, depth registration was via a laser (2018) or a cable encoder (2019). In the 2019 campaign the sticks were melted as is; in 2018, the sticks were cut into 50 cm sections to fit the smaller holder frames. The melt head barrier (26 x 26 mm) ensured separation of the outer (potentially contaminated) sample stream from the inner (clean) melt stream during melting. The inner melt stream was continuously

pumped through a buoyancy debubbler to remove air bubbles (see Bigler et al. (2011)).

The debubbled melt stream was split into several analytical systems, including insoluble dust (Abakus), conductivity and several simple fluorescence and absorption methods to analyse ammonium ($NH_4^+$), Sodium ($Na^+$) and calcium ($Ca^{2+}$) (Bigler et al., 2011), hydrogen peroxide ($H_2O_2$) and acid ($H^+$) (Kjær et al., 2016, 2022). Chemical signals were converted to concentrations using a linear regression produced by a set of two ($H_2O_2$ and acid) or three ($NH_4^+$, $Na^+$ and $Ca^{2+}$) known standards,

and the measured flow rate was used to convert the dust concentrations from counts per second to particles $ml^{-1}$. Calibrations were performed before and after each sample run (approximately 4 hourly). Around 5 m (2018) or 10 meters (2019) of ice were melted sequentially, prior to returning the system to ultrapure de-ionised water (Milli Q) to establish a baseline and run standards. At the start and end of each sample run, system blanks of de-ionised ultrapure water were analysed. Melt speed was maintained at around 3 cm $min^{-1}$ (2018) or 3.5 cm $min^{-1}$ (2019) providing slightly higher resolution in the 2018 data.



An additional line of debubbled water for the analysis of water stable isotopes ($\delta^{18}$O, $\delta$D) was included for the 2019 CFA campaign analysing the wet drilled cores. The water isotope line was modified from the one described in Gkinis et al. (2010, 2011). Water was pumped through the system at a rate of 0.4 mL min$^{-1}$, with the rate controlled by a peristaltic pump. The water sample passed through a 2 μm filter to remove any particulate matter prior to injection into an evaporation oven (set at 170° C) via a 40 μm fused silica capillary). The evaporated sample was mixed with dry carrier air and transferred to the

CRDS optical cavity of a Picarro L2130-i at a humidity level of 15,000 ppm. In-house standards (calibrated to Vienna Standard Mean Ocean Water (VSMOW) and Standard Light Antarctic Precipitation (SLAP)) were used to calibrate the samples. The measurement noise as inferred by the integration of the power spectral density of the $\delta^{18}$O and the $\delta$D signals is 0.08 and 0.43 ‰ respectively. The mean accuracy of the record, calculated using a "check" water standard is 0.03 and 0.3 ‰ for $\delta^{18}$O and $\delta$D respectively. A detailed description of the CFA water isotope dataset is in preparation.

**2.5   Inductively coupled plasma sector field mass spectrometry of the halogens sodium, calcium, iodine and bromine**

    Discrete subsamples taken every 25 cm from the Copenhagen CFA campaigns were transported frozen to the Ca' Foscari University (Venice, Italy) and stored at -20° C. Total Na, Ca, bromine (Br) and iodine (I) concentrations were determined with an Inductively Coupled Plasma Sector Field Mass Spectrometer (ICP-SFMS, Thermo Scientific™ ELEMENT2™) equipped with a cyclonic spray chamber. The discrete samples were not acidified to avoid possible volatilization of I and Br during the

analytical sequence. A ten-hour cleaning sequence was run before the sample analytical sequence, by alternating ultra-pure water and a solution of 2% ultra-pure HNO$_3$, until the blanks reached low and stable levels. I and Br analyses are susceptible to instrumental memory effects (Vallelonga et al., 2021). Given the low average concentrations determined for halogens in ice cores and the halogen memory effect, it is essential to ensure minimum background conditions between each sample analysis, thus the ICP-SFMS was cleaned between samples by drawing 2% HNO$_3$ and then de-ionised ultra-pure water for 60 seconds

each respectively. Flowing ultra-pure water through the ICP-MS tubes before the sample analysis has two effects. First, it removes any nitrate salts that may be deposited by the nitric acid and second, it conditions the instrument before introducing the unacidified samples.

    Concentrations were calculated by comparison to both external standards and blank-correction. External standards were prepared from mono-elemental solutions (TraceCert®, purity grade, Sigma-Aldrich, MO, USA) with concentration ranges

of 5 to 300 ng g$^{-1}$ (Na and Ca), 0.5 to 3 ng g$^{-1}$ (Br) and 0.05 to 0.3 ng g$^{-1}$ (I). The calibration curves showed a linear relationship and had a correlation coefficient greater than 0.99 (p-value ≤ 0.05). Blanks were measured every 15 samples and about 10% of the samples were re-analyzed to ensure repeatability. The limit of detection (LOD), calculated as 3.3 times the standard deviation of the blanks, was 0.1 ng g$^{-1}$ (Na), 3.6 ng g$^{-1}$ (Ca), 0.05 ng g$^{-1}$ (Br) and 0.002 ng g$^{-1}$ (I). The repeated samples showed a variability of 2.1% (Na), 4.5% (Ca), 3.1% (Br) and 7.7% (I) with respect to the original measurement. The

paleoenvironmental value of this dataset from the MBS site is under investigation, however for the purposes of this study, we present the annual mean halogen concentrations at MBS compared to those from the Law Dome ice core.



## 2.6 Imaging of stratigraphic features via intermediate layer core scanning (ILCS)

The MBS ice cores contain numerous stratigraphic features, particularly thin crusts. These crusts are qualitatively similar to bubble free layers identified elsewhere in Antarctica and Greenland, and they are distinct from melt layers in appearance by being thin (1-2 mm thick) and sharply defined rather than thick and ragged in appearance. In the WAIS Divide ice cores, such bubble free layers have been determined to be unrelated to melt events via noble gas analysis (Orsi et al., 2015). The MBS bubble-free layers appear regularly, with multiple instances per annual layer. In order to investigate these features and any climate information they may hold, stratigraphic imaging of the remaining half sections of all MBS ice cores after glaciochemical sampling was undertaken using an Intermediate Layer Ice Core Scanner (ILCS, Shåfter&Kirchoff™) and image acquisition system. For the purposes of this study, the ILCS images were used to help align depths across different samples and laboratories due to the significant depth alignment issues we encountered (see below), thus we describe the processing of ice core slabs for ILCS image acquisition here. The climatological or physical causes of these bubble free layers in coastal East Antarctic cores and their variability through time are currently under investigation elsewhere (Zhang et al., 2023). An example of an MBS core ILCS image used to constrain the depth scale is given in Appendix A1.

ILCS is commonly used to image ice cores for analysis or dating of dust or volcanic layers, generally in deeper ice cores drilled from below bubble close-off depths (Svensson et al., 2005; McGwire et al., 2008; Winstrup et al., 2012; Westhoff et al., 2022). The ILCS instrument consists of a frame surrounding a removable carriage which holds the polished ice core slab to be imaged. A camera mounted above the carriage and an indirect light source mounted below at an angle of 45° relative to the ice move synchronously from one end of the ice core slab to the other. The image acquired reveals clear ice appearing black, with impurities and bubbles in the ice on a greyscale.

Pre-processing of the ice core surface to be scanned was necessary in order to avoid interference from surface artefacts of core processing (e.g. bandsaw blade marks). We used a modified and cleaned timber planer/thicknesser modelled on a simplified version of the set-up used at the National Ice Core Facility (Colorado, USA) (McGwire et al., 2008). We designed a polypropylene tray system that allowed ice core slabs of varying dimensions to be secured safely in place while passing the tray through the thicknesser. Swarf produced during planing was removed with a clean, commercial vacuum cleaner attached to the dust port of the planer. We also used clean brushes and scrapers similar to those used in our ice core processing freezer to remove any particles of ice or dust remaining after planing that would interfere with image acquisition.

Once planed and microtomed to a smooth surface, the core slabs were left overnight to sublimate slightly which allowed a polished surface to develop. Each core slab was then scanned multiple times at different focus heights and brightness settings to ensure optimum image quality. We also used multiple changes in focus heights and brightness settings to acquire images of firn cores that would preserve features such as bubble free layers and crusts in shallower records (Zhang et al., 2023).

## 2.7 Constructing the MBS depth model

Ideally, the top and bottom depths and individual lengths of ice cores align perfectly to those measured during drilling, yielding insignificant discrepancies between core and sample lengths between laboratories or sampling campaigns. In reality this is not



always the case. We found instances where the length of MBS cores during sample cutting and pre-processing for analysis
was different to the field camp lengths. Mostly this appeared due to slanted core ends (see Appendix A1) from imperfect
cuts during field processing, as only hand saws were available for post-drilling processing. Additionally, two transport boxes
containing MBS Main cores (12 metres of ice) experienced major damage during transit from the field site to the Hobart
laboratories. These broken cores were difficult to re-assemble, and in one instance resulted in a length discrepancy of 1.5cm.

Overall, discrepancies between field and laboratory core lengths were usually less than 0.5 cm, but in rare cases up to 1.0 cm
or even 1.5 cm. Note that these field to lab discrepancies are similar to those found in other ice core studies, for example, see
Erhardt et al. (2022).

To solve the length discrepancies and derive a master depth model, we compared field and lab measurements to the ILCS
scans to derive the correct length of each core. This painstaking process was then used to determine the full drilled depth

of 294.785 metres, which is 18.5 cm different from the field measured depth of 294.6 metres (i.e. a 0.003% error). This
suggests that despite the discrepancies between field and laboratory measurements, the ultimate depth of the core was measured
accurately and individual core discrepancies will have limited impact on sample-depth registration. For the CFA analysis of
trace impurities and water isotopes, a process of scaling and shift factors was derived to account for length discrepancies. This
was necessary as the length of the CFA analysis sticks sent to the University of Copenhagen were sometimes shorter or longer

(by up to 0.5 cm) than the lengths of the original cores, depending on their orientation to any slanted end cuts in the whole core.
The scaling and shift factors will be described in detail as the CFA trace chemistry and water isotope datasets are developed
and published. For the predominantly discrete analyses used for the dating described herein, the depth model derived has been
applied to each sample.

## 2.8 Dating the Mount Brown South Main ice core

We followed an independent plus final consensus approach to dating using the seasonally varying concentrations of multiple
trace chemical species and water isotopes for identification of annual layers. An initial study of the satellite era of the MBS
Main and surface cores identified that annual horizons adjacent to low accumulation years could be difficult to detect due to
truncated or unclear seasonal cycles, a finding that was reinforced by comparison with modelled precipitation and surface mass
balance data for the MBS site (e.g. see Figure 2, Crockart et al. (2021)). With this in mind, we developed a methodology to

produce the MBS Main core chronology and revisit the age-depth scales derived for the surface cores and satellite era portion
of the Main core, as well as quantifying uncertainty from difficult to detect annual horizons through time.

A 'working' age-by-depth scale (hereafter $MBS_{CP}$) was developed using a layer counting approach similar to that derived
for Law Dome (Jong et al., 2022; Vance et al., 2022; Roberts et al., 2015; Plummer et al., 2012). $MBS_{CP}$ relied on sea salt
minima, isotope maxima, $nssSO_4^{2-}$ maxima and the summer peak in the sulfate to chloride ratio to identify annual horizons,

which were assigned a date of 1 January. This is different to Law Dome, where multiple analyses using co-located automatic
weather station data compared to ice core isotope records have determined the annual horizon to have a mean date of 10
January (van Ommen and Morgan, 1997; Jong et al., 2022). Constraining the mean date of the annual horizon to this degree
is unusual because it requires relatively high and uniform annual accumulation rates and long-term co-located meteorological





instrumentation at the site, which is not currently possible at MBS. However, we wish to remind future users of both the

MBS and Law Dome records of the difference in the actual date signified by the annual horizons in both records. Preliminary alignment to known volcanic ties for Antarctic ice cores was also undertaken via comparison of $nssSO_4^{2-}$ peak occurrence, shape and relative concentration.

Additionally, a layer counted only age-by-depth scale - $MBS_{HK}$ - was produced without reference to $MBS_{CP}$ by a separate investigator on this project. $MBS_{HK}$ used similar species to determine annual horizons to $MBS_{CP}$ with one clear difference;

this investigator identified a possible minimum in fluoride concentrations adjacent to the annual horizon of other species and hypothesised fluoride at MBS may follow a seasonal cycle. The investigator deriving $MBS_{HK}$ did not attempt to locate or incorporate known volcanic age ties, in order to deliberately derive a layer-counted only age scale that could be used to assess how frequently annual layers were 'missed' due to either a different dating approach and/or poor seasonality. Both these interim age scales defined three types of annual layers – certain years, uncertain but counted years (where there is deemed to be an

annual horizon but its exact placement is uncertain) and uncertain uncounted years (where there is some evidence for an annual horizon but not enough to include in the age scale).

We then formed the dating team (comprising the bulk of the authorship of this study) to incorporate the interim age scales toward a final chronology. We examined $MBS_{CP}$ and $MBS_{HK}$ on the same depth scale (see Appendix B1 for an example section). If $MBS_{CP}$ and $MBS_{HK}$ showed agreement on annual horizons, no change was made. Where horizons differed in

number or position, the final decision of annual horizon placement was made by group consensus and applied to $MBS_{CP}$. Finally, the group consensus version of $MBS_{CP}$ was synchronised to the West Antarctic Ice Sheet (WAIS) Divide 'WD2014' chronology (Sigl et al., 2016) to produce the chronology described here - $MBS_{2023}$. In $MBS_{2023}$, uncertain years are defined as counted or uncounted as above.

## 3 Results

### 3.1 $MBS_{2023}$ - a layer-counted chronology for the MBS ice cores

Our dating approach allowed for the comparison of independent plus consensus efforts by researchers skilled in dating ice cores from Antarctica and Greenland, and incorporated a range of experience in the final group assessment of annual horizon category and placement. In addition, we re-examined the dating for the satellite era portion of Main and of Alpha, Bravo and Charlie from Crockart et al. (2021). This re-examination did not result in any changes to the original dating, thus the analytical

time series and findings detailed in Crockart et al. (2021) and Jackson et al. (2022) remain current and are upheld. The time periods covered by the three surface cores are 1977-2017 (Alpha), 1978-2017 (Bravo) and 1965-2017 (Charlie).

For MBS Main, we found the 4.265 - 294.785 metre deep record spanned 873-2009 CE (1,137 years), fulfilling the stipulations of the site selection of a millennium-length record. For the majority of MBS Main, annual horizons were relatively easy to discern, but there were periods where seasonal cycles were less pronounced and it was more difficult to define the number

or placement of annual horizons. In total, we found 1,004 annual horizons to be certain counted years (i.e. the dating team deemed both the evidence pointing to the existence of an annual horizon as well as its exact placement on the depth scale to be





easy to discern). The remaining 133 horizons (11.7%) were uncertain counted years. Finally, we also found 73 instances across the Main record of what we deemed uncertain uncounted years, where there was only partial evidence of an annual horizon. An example would be a minimum in sea salt concentrations that was not matched by a corresponding peak in either isotopes

or the sulphate:chloride ratio. To reiterate, an uncertain *counted* year constitutes the *definite existence of a year, but it is unclear exactly where the annual horizon signifying 1 January should be placed*, while an uncertain *uncounted* year constitutes *incomplete evidence for a year that is not compelling enough to be included in the chronology*.

As with other more coastal Antarctic ice core records, volcanic horizons in the MBS ice core (deviations from the $nssSO_4{}^{2-}$ background) can be noisier signals compared to inland records due to the dilution of the signal in coastal regions (Plummer

et al., 2012). Despite this, the volcanic age ties we identified were clear enough to be very useful in evaluating the uncertainty that unclear annual horizons introduces to the $MBS_{2023}$ chronology (Table 3 and Figure 4). We compared and contrasted the likely volcanic signals we discerned in the chemistry record to those of Law Dome (1,377 m a.s.l., lower elevation than MBS) (Jong et al., 2022) and the Roosevelt Island Ice Core (79.36 , 161.71 W, 550 m a.s.l., lower elevation than both Law Dome and MBS) (Winstrup et al., 2019). We also tied the MBS record to the established West Antarctic Ice Sheet (WAIS) Divide

chronology, WD2014 (Sigl et al., 2016). Of note is the higher number of matched volcanic horizons in MBS to both the WAIS Divide and Law Dome records compared to the Roosevelt Island record. As Roosevelt Island is a much lower elevation record, the influence of biogenic sulfur made identifying small or short lived volcanic events difficult, leading to fewer matched horizons (Winstrup et al., 2019). Below 256.46 metres depth (1,040 CE), the $MBS_{2023}$ chronology is layer counted only and not synchronised to WD2014 due to the lack of an identifiable volcanic horizon below this point.

**3.2 Mean annual concentrations and seasonal cycles of key trace chemical species**

Mean annual concentrations of key trace chemical species at MBS are generally lower than those reported from Law Dome, except for nitrate and methanesulfonic acid (Table 4). We also examined seasonal cycles for sodium, non-sea-salt-sulfate, the sulfate to chloride ratio and fluoride (Figure 5). As is commonly reported, this type of analysis is reliant on an assumption of uniform accumulation through the year at the site due to the lack of within year dating markers (Winstrup et al., 2019;

Kjær et al., 2022). Uniform accumulation is not realistic, and we know MBS has a precipitation bias with lower summer accumulation during the satellite era (Crockart et al., 2021; Jackson et al., 2022). Nonetheless, seasonal cycle analysis is useful for examining the relative timing of minima and maxima between individual species at a given ice core site.

The MBS fluoride record presented here is an interim dataset, with some missing sections due to variability resulting from the very low concentrations present at MBS and possible losses in the firn section during storage. We present fluoride seasonal

cycles covering time periods of adequate sample fidelity from the MBS Main and Charlie cores (Figure 5). These show that fluoride has a minimum in early summer (November/December) and an autumn maximum (March/April/May). Note that the satellite era fluoride seasonal cycle was derived without reference to fluoride as an indicator of an annual horizon, as the satellite era age scales were developed in Crockart et al. (2021) prior to examination of the fluoride record.





**Table 3.** Proposed volcanic dates and depths from the MBS ice core. The MBS record has been synchronised to WAIS Divide (Sigl et al., 2016) at the year level, thus WAIS errors should be taken into account. Note however, intra-year synchronisation has not been done, to illustrate differences in the onset of the volcanic signal. Comparisons with Law Dome (Plummer et al., 2012; Jong et al., 2022) and Roosevelt Island (Winstrup et al., 2019) are also shown. Note the Kuwae date may be disputed depending on which peak is matched in the 1450s (see Plummer et al. (2012), however, this is the equivalent peak matched by peak size.

| Proposed Event | MBS Depth (m) | MBS | WAIS Divide | Law Dome | Roosevelt Island |
|---|---|---|---|---|---|
| Pinatubo | 12.50 | 1992.9 | 1993.0±0 | 1992.8±0 | Not matched |
| Agung | 25.39 | 1965.1 | 1965.1±0 | 1965.1±0 | Not matched |
| Krakatoa | 53.96 | 1885.0 | 1885.0±1 | 1885.0±0 | 1885.0±1 |
| Makian | 59.67 | 1864.0 | 1863.9±1 | Not matched | 1863.3±3 |
| Cosiguina | 67.39 | 1837.0 | 1836.8±1 | 1836.9±0 | Not matched |
| Unknown | 68.55 | 1832.0 | 1832.1±1 | 1832.1±0 | Not matched |
| Galunggung | 70.91 | 1824.3 | Not matched | 1824.1±0 | Not matched |
| Tambora | 73.24 | 1816.3 | 1816.4±0 | 1816.3±0 | 1818.0±5 |
| Unknown | 74.88 | 1810.3 | Not matched | 1810.3±0 | Not matched |
| Unknown | 86.49 | 1763.2 | 1762.8±1 | 1762.9+1 | Not matched |
| Unknown | 104.32 | 1695.9 | 1695.8±1 | 1695.9+1 | 1695.0±8 |
| Gamkonora | 109.15 | 1675.0 | Not matched | 1675.2+1 | Not matched |
| Unknown | 114.09 | 1655.5 | 1655.5±1 | 1655.4+1 | Not matched |
| Parker Peak | 117.83 | 1642.4 | 1642.4±1 | 1642.6+1 | 1641.2±8 |
| Unknown | 123.72 | 1621.1 | Not matched | 1621.2+1 | Not matched |
| Huaynaputina | 129.24 | 1601.6 | 1601.4±1 | 1601.5+1 | 1599.3±9 |
| Ruiz | 131.15 | 1596.0 | 1596.0±1 | 1596.0+1 | Not matched |
| Kuwae | 162.96 | 1459.4 | 1459.8±2 | 1459.7+1 | 1458.4±11 |
| Unknown | 180.79 | 1390.0 | Not matched | 1390.0+1 | Not matched |
| Unknown | 182.58 | 1382.1 | 1381.7±2 | 1382.0+1 | Not matched |
| Unknown | 192.02 | 1345.8 | 1345.4±2 | 1345.0+1 | Not matched |
| Unknown | 205.02 | 1287.3 | 1286.8±2 | 1286.5+1 | Not matched |
| Unknown | 206.88 | 1277.5 | 1277.2±2 | 1276.6+1 | 1277.3±13 |
| Unknown | 208.46 | 1269.5 | 1269.7±2 | 1268.8+1 | 1269.9±13 |
| Samalas | 210.46 | 1258.3 | 1258.9±1 | 1258.3+1 | 1257.3±13 |
| Unknown | 213.58 | 1241.9 | 1241.9±2 | Not matched | 1242.3±13 |
| Unknown | 215.64 | 1230.8 | 1230.7±2 | 1230.0+1 | 1231.4±14 |
| Unknown | 223.55 | 1192.0 | 1191.9±2 | 1191.3+1 | 1190.1±17 |
| Unknown | 227.58 | 1172.3 | 1172.4±2 | 1171.4+1 | Not matched |
| Unknown | 240.23 | 1110.2 | 1110.1±2 | Not matched | Not matched |
| Unknown | 246.36 | 1082.2 | 1082.0±2 | Not matched | Not matched |
| Unknown | 255.58 | 1040.6 | 1040.3±2 | Not matched | 1043.3±19 |





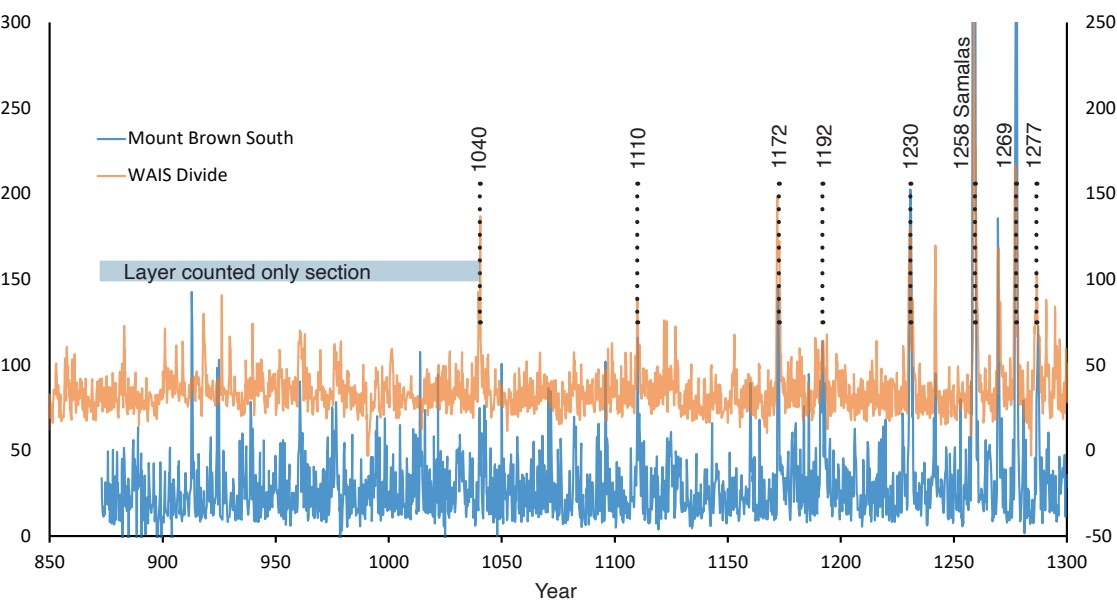

**Figure 4.** The nssSO$_4$$^{2-}$ record (ppb, left axis) from the deepest third of the MBS Main core (202.4-294.6 m) synchronised to the WAIS Divide non-sea-salt sulfur record (ppb, right axis, offset for visual clarity) showing volcanic tie-points used to synchronise WAIS Divide and MBS. Refer to Table 3 for the full list of tie points used for synchronisation and comparison). The end of the synchronised chronology and commencement of the layer-counted only chronology for the MBS Main record at 1040 CE to the bottom of the core is indicated.

## 4 Discussion

We present chronologies for a new multi-century and three surface ice cores from the boundary of Wilhelm II and Princess Elizabeth Land in East Antarctica. The ice core site - Mount Brown South - was drilled in 2017/2018 and spans 873-2009 CE and up to 2017 CE with the inclusion of the surface cores. The chronologies developed here - MBS$_{2023}$ - incorporated expertise from an international team with experience in dating both Antarctic and Greenland ice cores.

### 4.1 Seasonal cycles of trace chemical species and the potential of fluoride as a climate proxy at Mount Brown South

The MBS records display clear seasonality in three primary analytes that are frequently used for layer counting - sea salts (principally sodium and chloride), non-sea-salt-sulfate and the ratio of sulfate to chloride (Figure 5). Seasonal cycles of sea salt and non-sea-salt-sulfate exhibit expected maxima in winter (June-September) and summer (December-February) respectively, leading to a summer peak in the ratio of sulfate to chloride. We define the MBS annual horizon to have a nominal date of 1 January. The installation of automatic weather stations at an ice core site over multiple years (e.g. as is the case at the Law 350 Dome ice core site) can constrain the mean date of the annual horizon in a site specific way, e.g. see van Ommen and Morgan (1997). This would be difficult to achieve at MBS given the lack of co-located instrumentation and the lower and more episodic





**Table 4.** Mean annual concentrations of key trace chemical species from the MBS Main core from 873-2009 CE compared to the Law Dome ice core from 1-2017 CE (Jong et al., 2022) except for methanesulfonic acid (1750-1995 CE) (Curran et al., 2003), bromine and iodine (1927-1989 CE) (Vallelonga et al., 2017). The fluoride mean concentration at MBS and limit of detection is an interim calculation determined using the method of Sanz Rodriguez et al. (2019). Analytical methods for ion chromatography are according to Sanz Rodriguez et al. (2019) and Curran and Palmer (2001), and publications for the source data for the comparison are indicated.

| Species | MBS (ppb ± 95% CI) | Law Dome (ppb) | Limit of detection | Analytical method |
|---|---|---|---|---|
| Methanesulfonic Acid | 7.4 ± 0.97 | 4.2[a] | 0.01[a] | IC |
| Fluoride | 0.13 ± 0.038 | Not reported | 0.008 | IC |
| Chloride | 38.6 ± 15.2 | 150.7[b] | 0.21[b] | IC |
| Sodium | 18.4 ± 7.6 | 82.30[b] | 0.23[b] | IC |
| Nitrate | 48.98 ± 9.92 | 22.94[b] | 0.31[b] | IC |
| Sulfate | 32.66 ± 9.66 | 36.98[b] | 0.38[b] | IC |
| Potassium | 3.13 ± 0.78 | 11.73[a] | 0.39[a] | IC |
| Magnesium | 2.19 ± 0.73 | 10.09[b] | 0.49[b] | IC |
| Calcium | 4.41 ± 2.81 | 9.62[a] | 0.40[a] | IC |
| Bromine | 0.254 ± 0.008 | 5.15[c] | 0.06[c] | ICP-MS |
| Iodine | 0.00045 ± 0.0005 | 0.0639[c] | 0.0002[c] | ICP-MS |

[a]Curran et al. (1998) [b]Jong et al. (2022) [c]Vallelonga et al. (2017)

accumulation regime. Some evidence for this is that the Charlie surface core shows slight differences in the annual horizon peak shapes compared to the Main core (Figure 5).

Fluoride is an infrequently investigated species in polar ice core studies, however it has been known to comprise part of the
trace chemistry suite of Antarctic ice cores for some time (Severi et al., 2014; Morganti et al., 2007). It is usually present in low concentrations and has been shown to have multiple sources, including anthropogenic production, sea salts, volcanoes, dust, and perhaps forest fires (De Angelis and Legrand, 1994; Preunkert et al., 2001). This study shows fluoride varies seasonally at MBS with a peak in austral autumn. We derived fluoride seasonal cycles for both the satellite era and in the pre-instrumental era. The satellite era layer counting of the Main and Charlie cores was derived without reference to fluoride as a potentially
seasonally varying species (as these sections were dated prior to any investigation of fluoride seasonality, and their ages at depth are unchanged in this study). The similarity in seasonal cycle shape between the pre-instrumental era in MBS-main compared to the satellite era periods in both MBS-Main and Charlie suggests a robust seasonal cycle is present.

Fluoride peaks in March/April/May (austral autumn) at MBS, neither coincident with sea salt sodium that peaks later in winter (June-September) nor non-sea-salt sulfate, which peaks in summer (probably related to algal blooms and production of
MSA). Thus, fluoride at MBS seems neither related to sea salt aerosol deposition nor biogenic activity. The fluoride minimum occurs during early summer (November/December) before the sea salt minimum in January, and approximately one month prior to the peak in the sulfate to chloride ratio, providing a potentially independent marker of the onset of summer at the MBS site.





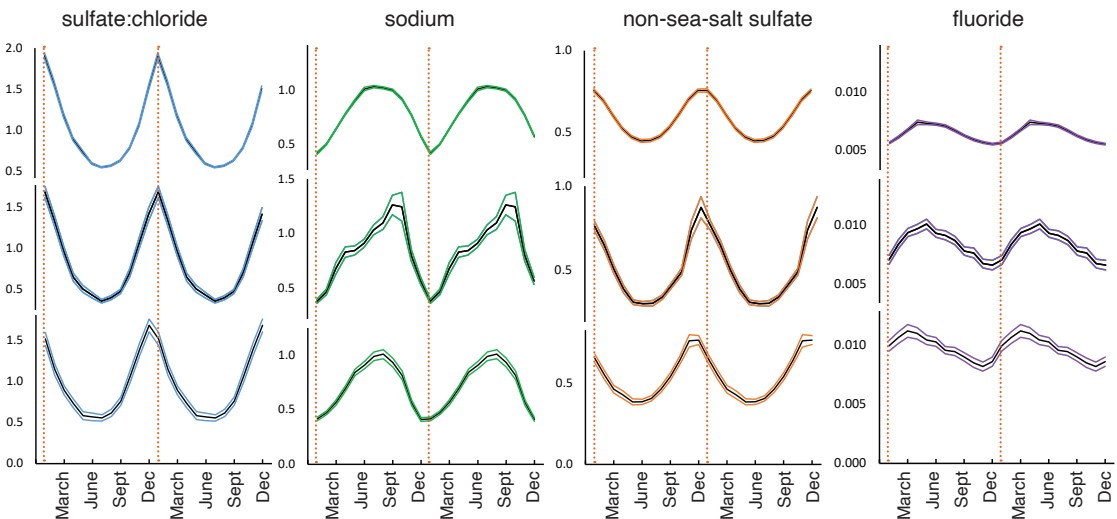

**Figure 5.** Seasonal cycles of key trace chemical species in the MBS ice cores. The sulfate to chloride ratio (blue), sodium (green) and non-sea-salt sulfate (orange) concentrations were the primary indicators of annual layers, with fluoride (purple) being a confirmatory species. Concentrations are shown in the units used during layer counting ($\mu$Eq L$^{-1}$). From top to bottom of each row are seasonal cycles for the Main core non-satellite era (872-1978 CE), the Main core satellite era (1978-2008 CE) and the Charlie surface core (1966-2017 CE). For fluoride, seasonal cycles were calculated over 962-1798 CE (top), 1978-2008 (middle) and 1966-2012 (Charlie) due to an incomplete dataset available to this study. The black line in each panel represents the mean value for that month for the time period selected, assuming uniform accumulation. The coloured upper and lower bands are the standard error of the mean. For orientation, vertical orange dotted lines indicate the annual horizon date of 1 January. For detailed analysis of water stable isotope ratios, their relationship to seasonality and the influence of biases due to moisture intrusions, see Jackson et al. (2022).

We are unaware of any regional anthropogenic, forest fire or volcanic sources that would lead to the fluoride seasonality observed at MBS. This leaves only two possible sources from the literature. Fluoride could be deposited as part of the dust fraction at MBS (presumably from the Vestfold Hills, the only regional ice-free area that could provide a dust source of any magnitude). Or, there could be a solar driven loss cycle to the sea salt fraction from mid-winter (however the early maxima in March or April makes this hypothesis unlikely based on seasonal cycle shape).

A third possibility that has not been canvassed in the literature, but that should be investigated based on the shape and timing
of the seasonal cycle is that there is a relationship between fluoride deposition and sea ice seasonality, given Antarctic sea ice retreats to its minimum extent in February/March (Raphael and Handcock, 2022). The autumn peak in fluoride would suggest a source related specifically to sea ice formation and production (which may or may not be related to maximum extent). The





fluoride seasonal cycle at MBS displays an asymmetry analogous to that of Antarctic sea ice, which has a seven month growth phase and a five month decline phase (Eayrs et al., 2019; Roach et al., 2022).

A possible mechanism linking flouride seasonality at MBS and sea ice formation is the behaviour of East Antarctic polynyas. The fluoride maximum in April occurs when the sea ice production rate in the Cape Darnley Polynya (on the western edge of the Amery Ice Shelf around 800 km from the MBS site) is at a maximum. The polynya reaches its maximum size in March and again in November (Fraser et al., 2019). However the sea ice production peak is in April, and is related to the March maximum, where the export of newly formed sea ice in the polynya occurs at a time when the polynya size is not constrained by the presence of a consolidated ice pack to the north. That is, the ice tongue to the north of the polynya has not yet developed in April, so the ice formed is more easily exported and the polynya size is large (Fraser et al., 2019). Polynya size is also large in November, but this is de-coupled from sea ice production. This climatological cycle of sea ice production related to polynya size is likely to be common to many polynya areas across East Antarctica, not just Cape Darnley. The Cape Darnley Polynya is regionally significant to sea ice production, but it also has global impacts on thermohaline circulation as it is the second highest area of Antarctic Bottom Water formation after the Ross Ice Shelf Polynya (Ohshima et al., 2013). A proxy of sea ice production in polynyas would be of huge value, and is worthy of future investigation particularly given recent developments in the ability to discern between land fast ice, frazil ice and newly consolidated sea ice (Fraser et al., 2021; Nakata et al., 2021). Nonetheless, any relationship to sea ice formation would need to be disentangled from any regional dust sources (e.g. the Vestfold Hills) - the only other likely significant source leading to the observed seasonal pattern at MBS. In addition, a closer examination of other known sea ice proxies (e.g. MSA and the halogens) at MBS is warranted.

    While the seasonality of the fluoride record from MBS may represent a climatological link, its low concentrations make measurement challenging (Sanz Rodriguez et al., 2019) and the reliability of the record is variable at different periods of the MBS record. In addition, its volatile nature means it is subject to post-depositional modification (Legrand et al., 1996). This means development and interpretation of the fluoride record will need to proceed with caution, and the possible competing sources in the MBS/East Antarctic region quantitatively explored.

### 4.2 Annual mean concentrations of trace chemical species

The mean annual concentrations of the sea salts sodium, chloride, potassium and magnesium at MBS occur at around a quarter of the concentrations at Law Dome (Table 3). Sea salts have short atmospheric residence times of at most a few days (Schüpbach et al., 2018), suggesting the mean trajectory of sea salt aerosol reaching MBS has a less immediate link to maritime airmasses than at Law Dome, resulting in relative depletion. This suggests that while MBS is a coastal core geographically, climatologically it is a more inland site than Law Dome. The MBS site is approximately 700 m higher than Law Dome and is thus at a transition zone to plateau records. The depleted concentrations of calcium compared to Law Dome (by around 50%) are probably similarly related to more continental trajectories occurring at MBS, although the atmospheric residence and transport times of calcium can be longer than sea salts (Schüpbach et al., 2018).

In contrast, sulfate concentrations at MBS are similar to Law Dome, while mean annual concentrations of methanesulfonic acid (MSA) are around double those of Law Dome. This may suggest higher regional production of MSA close to MBS com-





pared to Law Dome. MBS is relatively close to the Vestfold Hills, one of the largest ice free areas of East Antarctica with a number of field stations including Davis station (approximately 380 km to the west). Numerous studies of coastal phytoplankton production and seasonality have been conducted at Davis station, including studies of the phytoplankton synthesised

precursor of MSA, dimethylsulfide (DMS), and its biochemical precursor dimethylsulfoniopropionate (DMSP). These studies have detailed exceptionally high seasonal concentrations of DMS and DMSP in late season sea ice, and during and after sea ice breakout (Trevena et al., 2000, 2003; Trevena and Jones, 2006). The high concentrations may be related to rapid sea ice breakout in combination with fertilization due to wind-blown dust from the Vestfold Hills. This would elicit rapid phytoplankton growth and the production of DMS and DMSP from increases in sulfur-producing phytoplankton species (Vance et al., 2013).

However, there could be another MSA source further to the northwest of MBS. Air parcel trajectory studies show that MBS is regularly 'downstream' of the Kerguelen Plateau (Jackson et al., 2022), another region of episodically very high phytoplankton production (Robinson et al., 2016; Schallenberg et al., 2018). The high concentrations of MSA at the MBS site in combination with a prior study of an earlier MBS short core investigating MSA as a sea ice proxy, are promising for the development of new sea ice reconstructions from the MBS ice cores (Curran et al., 2003; Foster et al., 2006).

The halogens bromine and iodine show far lower concentrations at MBS with respect to Law Dome over the 20th century (Vallelonga et al., 2017). On average, bromine and iodine levels are at least 20 times lower with respect to Law Dome (in the case of iodine, two orders of magnitude lower). As noted previously MBS is located at a higher elevation than Law Dome, which may influence the amount of impurities delivered to the site (Bertler et al., 2005). However the low concentrations of halogens are curious given the coincident higher concentrations of MSA at MBS, as both MSA and halogens are proxies of sea

ice.

The differences in annual concentrations between the two sites may also be related to site specific factors and local climatology. Law Dome is a small semi-independent ice cap which is only around 100 km from the coast to the west, north and east. During summer this region is essentially sea ice free, and this proximity to the coast and the orographic nature of the dome leads to very high accumulation that is comparatively seasonally uniform as tropospheric moisture is advected onto the dome

directly off the ocean (Roberts et al., 2015; Crockart et al., 2021; Udy et al., 2021, 2022). MBS is not significantly further from the coast in a northerly direction, but in terms of its eastern prevailing flow, moisture reaching the MBS site may have to travel over continental East Antarctica for many 100's of kilometres (although this depends on the trajectory of the air mass, see Udy et al. (2022) and Jackson et al. (2022)). It is likely the more episodic precipitation at MBS combined with an intermittent continental source trajectory may contribute to the differing concentrations in chemical species.

### 4.3 Dating the Mount Brown South Main ice core

Sequential layer counting efforts of the annually resolved section of the Law Dome ice core (surface to 800 metres) over the last three decades assisted the development of $MBS_{2023}$ (e.g. (van Ommen and Morgan, 1997; Curran et al., 1998; Vance et al., 2022; Jong et al., 2022) and small but key differences during layer-counting at MBS compared to Law Dome were noticed. Primarily these differences concerned the relative clarity of the water stable isotope records compared to the trace chemistry records. The summer peak in $\delta^{18}O$ is generally the primary determinant of an annual layer at Law Dome, with trace chemical




species providing confirmatory evidence (van Ommen and Morgan, 1997; Plummer et al., 2012). In contrast, we found the evidence of annual layers in the sea salt, non-sea-salt-sulfate and fluoride concentrations and the sulfate to chloride ratio were equivalent to the $\delta^{18}O$ evidence at MBS, and considering a combination of all species was the best approach. Jackson et al. (2022) determined that temperature biases (and subsequent effects on water isotope ratios) are present in extreme precipitation

events at MBS, leading to isotopic maxima which are more related to the moisture source and its synoptic trajectory, and less to site temperature. Additionally, summertime snowfall accumulation at MBS is lower than during the polar winter (March/April to November) (Crockart et al., 2021; Jackson et al., 2022). Given the satellite era mean annual accumulation of 300 kg m$^2$ compared to 690 kg m$^2$ at Law Dome, and the tendency toward reduced accumulation during summer (Crockart et al., 2021), it is likely that in some years summertime accumulation may be too low to adequately resolve the isotope peak from our sampling

resolution. In contrast, the winter peak in sea salts during higher wintertime accumulation may be more easily discerned. A final key difference between MBS and Law Dome is the clear seasonality of fluoride concentrations at MBS, however this difference relies on the development of a robust fluoride dataset from Law Dome to substantiate differences between the two sites.

During layer counting, we observed periods of multiple years where annual horizons changed from consistently easy to

discern to periods with unclear horizons (see Appendix B1). These periods may be related to years with either lower than average snowfall accumulation, or higher than average episodic accumulation, resulting in an unclear seasonal cycle. Our approach using interim age scales allows the estimation of uncertainty between volcanic tie points as a result of uncertain or unidentified annual layers in the layer-counted only MBS$_{HK}$ (Figure 6). Crockart et al. (2021) hypothesised that low or no accumulation periods at MBS would lead to dating error due to the absence of clear seasonal cycles. However the frequency of

identifiable volcanic horizons ensures that we can constrain uncertain horizons between volcanic ties, improve understanding of the episodic nature of accumulation at MBS and potentially identify areas of possible temperature bias in the isotope signal.

Knowledge that the trace chemistry seasonal cycles deserved equal weighting in discerning annual horizons as the water isotope records became increasingly evident during the group dating process. Ultimately, a cumulative error of 72 years of uncounted years is accrued in layer counting in MBS$_{HK}$ compared to the synchronized MBS$_{2023}$, even with uncertain horizons

included (Figure 6). However, we think this error has some mitigating factors. Firstly, the greatest 'loss' of annual horizons (25 y) is accrued from the top of the core to the first major tie at Tambora (1816), a rate of 13 years per century. This suggests this was either a period of 'training' for the researcher who produced MBS$_{HK}$, or, the higher number of samples per annual layer in the firn section led to more (rather than less) issues in identifying individual annual horizons. A second reason for the greater error in the upper 200 years is the issues with fluoride analyses in the top 80 metres of the MBS Main record (see methods).

This meant fluoride was unavailable to assist with dating for this section. After this upper section, the error rate per century drops markedly to 3-6 years per century. We hypothesise that if a new MBS ice core was retrieved and a complete fluoride dataset available, the accumulated knowledge of the inherent idiosyncrasies of the site from this study would result in a much smaller error between the layer-counted effort and the final volcanic tie constrained chronology.

Curiously, from 1040 CE to the bottom of the core (approximately 40 metres), the error rate per century remains similar,

despite both efforts comprising layer counting only. We suggest that our four step dating approach of interim efforts followed



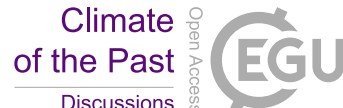

**Figure 6.** Assessing uncertainty from missed annual layers in the MBS ice core records. The upper panel shows the assigned annual layer date by depth between the final Main core chronology (MBS$_{2023}$, solid black line) compared to layer counting only (MBS$_{HK}$, orange solid and dotted lines). The layer counting efforts are further distinguished by whether uncertain uncounted annual layers were included or not (solid versus dotted orange lines respectively). The lower panel illustrates the resultant depth difference in metres for any given assigned annual layer date between the final chronology and the interim age-by-depth scales. Blue vertical lines link assigned annual layer dates at specific volcanic tie points (or the bottom depth at 873 CE / 295 metres) with their corresponding depth for MBS$_{2023}$, and link to the depth difference (in metres of ice core) shown in the lower panel. Within each tie point boundary the additional years difference between MBS$_{2023}$ to MBS$_{HK}$ is indicated, as well as the cumulative difference with increasing depth. For example, from the surface to the 1459 Kuwae horizon, there is a depth and age difference of 12.8 m / 47 y, which includes an additional 22 years from the previous tie point at 1816 (Tambora). On the bottom panel, the error rate (in years per century) is indicated for each tie point.





by group consensus is thus a robust approach to dating a record with the potential for low and/or episodic accumulation from one year to the next as it allows the independent development of age scale/s that can be compared with a final consensus scale to examine where error is being accrued. In this case, we think that the difference in the age scales from 1040 to the bottom of the core and the lack of a change in the slope of the age scales below 1040 CE indicates 'training' of the group to a new

site, which has resulted in the robust identification of unclear annual horizons after sequential reviews of the data, even in the absence of volcanic tie points.

### 4.4 Recommendations arising from this study

We did not make field drawings or take photographs of core orientation, stratigraphic features or drilling damage (e.g. core dog damage) during field processing. Such records would have provided additional physical records of core orientation and

features that would have helped solve discrepancies between core lengths measured in the field compared to in the laboratory. In addition, a Japanese Pull Saw and mitre box, or a mechanical drop saw in the field camp would have ensured consistent cuts at 90° to the plane of the core, and reduced subsequent errors encountered when constructing the depth model. Nonetheless, the ILCS images of the core we had access to were invaluable in aligning top and bottom depths, and we recommend ILCS image comparisons where possible to groundtruth depth models.

This study provides the individual chronologies (collectively MBS$_{2023}$) for all MBS cores drilled in 2017/2018 (Main, Alpha, Bravo and Charlie). Given the greater availability of spatial meteorological information since the satellite era in the Southern Hemisphere, users of the MBS ice core records may wish to develop records with as much overlap with the satellite era as possible. In this case, we suggest constructing a composite record using the Main and Charlie (which has a similar range of analyses to the Main core) chronologies to produce a single time series spanning 873-2017 CE. Our suggested compositing

point is the annual horizon of 1989/1990 (e.g. the 1 January 1990 horizon). This horizon has a clear and comparatively high peak in $\delta^{18}$O and a correspondingly low and clear summer sea salt minima. It is also prior to the nssSO$_4$$^{2-}$ signal attributed to the Pinatubo eruption (mid-1991) which can be discerned in all four records (See Figure 2, Crockart et al. (2021)).

### 5 Conclusions

Age-by-depth scales and preliminary results from the new Mount Brown South (MBS) ice cores from East Antarctica are

presented. We used a four step approach to develop chronologies for the 295 metre 'Main' core, and three 20-25 metre surface cores, via layer counting of seasonally varying species and alignment to known volcanic horizons. The MBS 'Main' ice core spans 1,137±2 years, and the three shallow cores span the recent 4-5 decades up to the surface age at the time of drilling (austral summer 2017/2018). Uncertainty in layer counting is assessed by comparing the final chronology with an interim layer-counted only age-by-depth scale to determine the rate of annual horizons missed due to muted seasonal cycles and/or

episodic snowfall accumulation. Detection of volcanic horizons is discussed and volcanic sulfate features are compared to other Antarctic ice cores. Mean annual concentrations and seasonal cycles of key trace chemistry species are derived and compared to the Law Dome record 1000 km to the east, with the concentrations of sea salt species and halogens generally less than a



quarter of those at Law Dome, probably due to the more continental transport on average at the MBS site. Conversely, concentrations of methanesulfonic acid are much higher than at Law Dome, which we attribute to higher local/regional phytoplankton productivity leading to increases in the biogenic precursors of MSA. Finally, we describe a distinct seasonal cycle in fluoride concentrations at MBS that may have a number of possible climatological links, including as a proxy for sea ice production in East Antarctic polynyas.

*Data availability.* Annual horizon depths and ages for all four MBS ice cores described here (Main, Alpha, Bravo and Charlie) will be available from the Australian Antarctic Data Centre upon publication.

## Appendix A: Example ILCS image

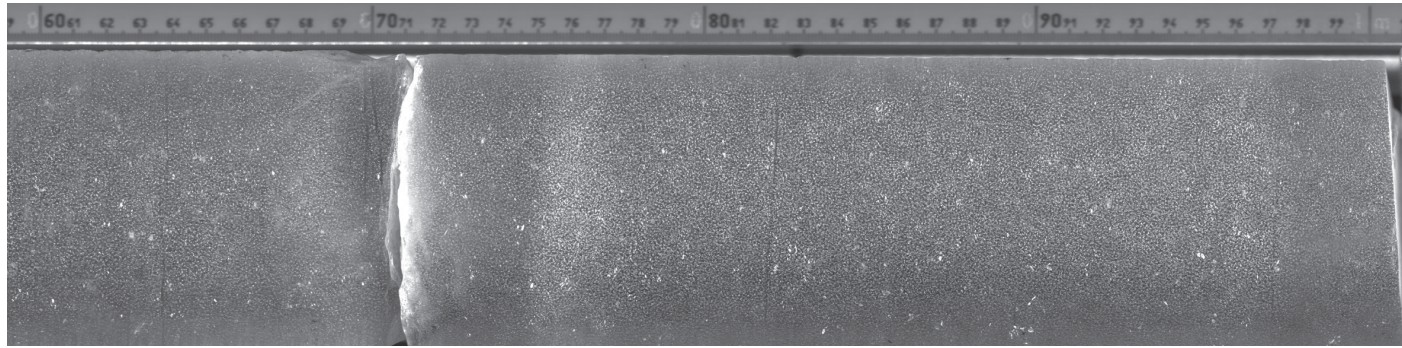

**Figure A1.** An example section of an Intermediate Layer Ice Core Scanning (ILCS) image from a deep MBS ice core (core 225, showing a section from 60-100cm of the core). A slanted cut at the end of the core can be discerned. The ILCS images were used to align and constrain differing core lengths between field and laboratory measurements. A core break (drilling break) can be seen at  70 cm.

## Appendix B: Annual layer counting example

*Author contributions.* TRV designed the study, led the writing with input from all authors, and led the Australian Antarctic Science logistics project to retrieve the ice core. TRV, NJA, ASC, CKC, ADeC, VG, MH, SJ, HAK, CAL, MN, CTP, DS, AS, PTV contributed to drilling, processing, analytical campaigns, production of the depth models, layer counting and data production. VF produced the site climatological data. CTP and HK produced interim age scales for comparison.



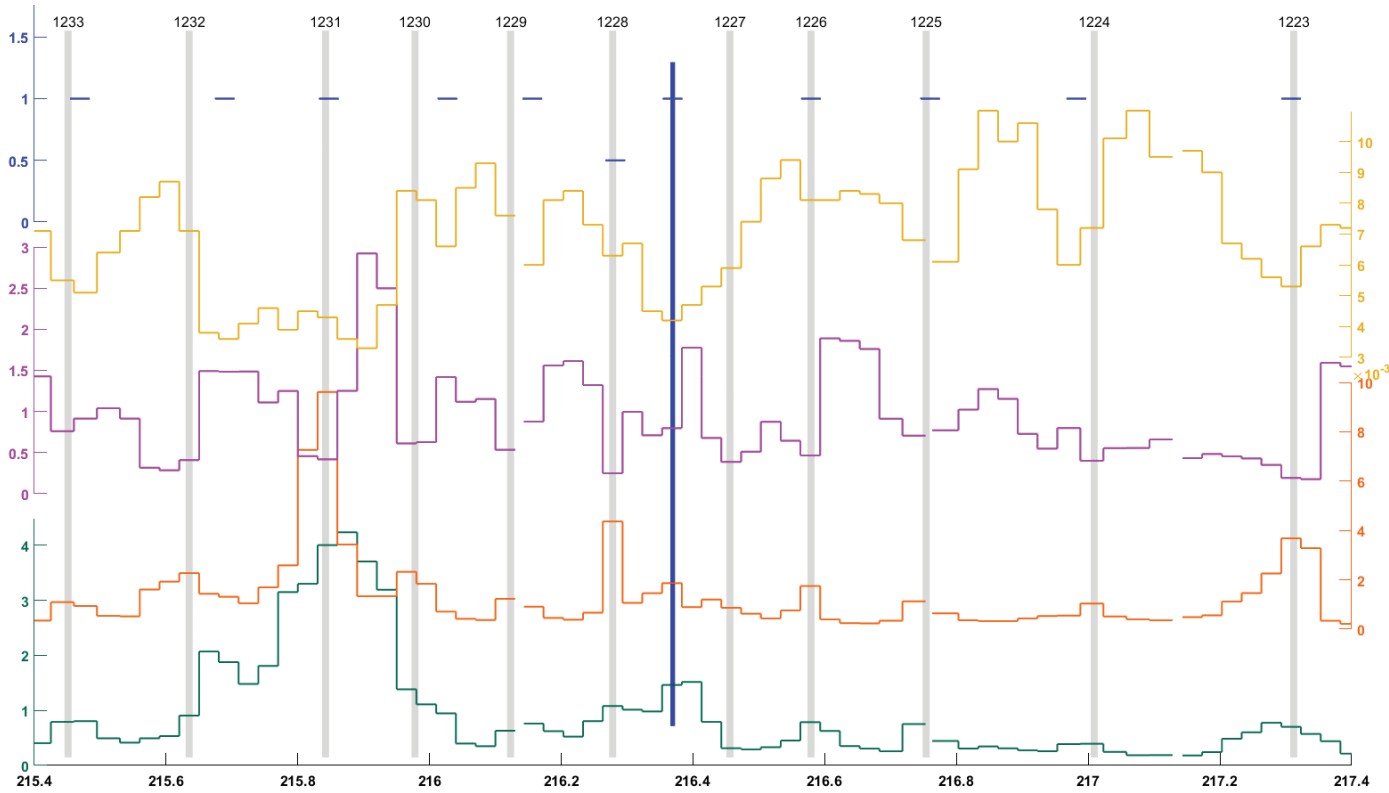

**Figure B1.** An example section of trace chemical data used to identify annual layers in the MBS Main ice core using the Matchmaker program. Ice core depth (m) is on the bottom x axis and the annual horizon placement and year assigned is shown at the top. Trace chemical species used are fluoride (yellow), chloride (pink), sulfate:chloride ratio (orange) and non-sea-salt-sulfate (green). Two different dating efforts are shown - the individual preliminary effort (MBS$_{HK}$) which has certain years shown as blue horizontal dashes, and uncertain years as blue dashes at a lower alignment. Vertical lines indicate the final consensus effort of MBS$_{2023}$, where grey lines represent certain years, bold blue lines represent uncertain counted years (not present in this example), and the thinner, shorter blue line is an example of an uncertain uncounted year. The commencement of a volcanic horizon (unknown) can be seen in the non-sea-salt sulfate data at 215.6 metres (see Table 3).



*Competing interests.* Nerilie J. Abram is a member of the editorial board of Climate of the Past.

*Acknowledgements.* We thank Jason Roberts, Sharon Labudda and Bloo Campbell for essential field work contributions, and Adam Treverrow for co-designing the ILCS laboratory and planer set-up.

*Financial Support* This study was supported by the Australian Government's Antarctic Science Collaboration Initiative (ASCI000002)
through funding to the Australian Antarctic Program Partnership. Logistics and analytical funding was provided by an Australian Antarctic Science grant (AAS 4414), the Australian Antarctic Division, the Carlsberg Foundation and a European Union Horizon 2020 research and innovation grant (TiPES, H2020 grant no. 820970). This study contributes to an Australian Research Council (ARC) Discovery Project (DP220100606) to TRV and NJA. NJA was supported by an ARC Future Fellowship (FT160102059), and NJA and SLJ were supported by the ARC Special Research Initiative Australian Centre of Excellence in Antarctic Science (SR200100008) and the Centre of Excellence
for Climate Extremes (CE170100023). ASC received support from Polar Knowledge Canada. VG acknowledges support from the Villum Foundation (project numbers 00022995, 00028061) and the Danish Independent Research Fund (DFF grant ID: 10.46540/2032-00228B). VF acknowledges support from Agence Nationale de la Recherche, projects ANR-20-CE01-0013 (ARCA), ANR-14-CE01-0001 (ASUMA).



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
