# Peer review of "An annually resolved chronology for the Mount Brown South ice cores, East Antarctica."

_Climate of the Past, 2023_

## Author Comment (AC1)

Author responses for Referee1 – Jacob Chalif

In the paper "An annually resolved chronology of the Mount Brown South ice cores, East Antarctica," T.R. Vance and coauthors present four new chronologies for ice cores (three surface cores and one deeper core) from Mount Brown South (MBS) in East Antarctica. These chronologies were developed through a combined multi-researcher annual chemical layer counting and volcanic alignment approach, which is common in dating annually resolved ice core records. Preliminary analyses of ice core chemistry, in particular comparisons with Law Dome and an analysis of the seasonal cycle of certain chemical species used in the annual layer counting effort, are presented as well.

I applaud the authors for their comprehensive discussion of the methods used in analyzing the MBS ice cores. Not all ice core studies are so transparent in detailing their methods of analysis, but the authors do a commendable job of laying out their complete analytical regime, which involved multiple cores, institutions, and types of chemical and physical analyses.

Given the quality of the annual layers in the MBS ice cores, I completely agree with the method the authors chose to use to establish the MBS chronologies.

In Section 4.1, the seasonal cycles of trace chemical species are discussed. This discussion is foundationally important for their dating methodology, but the authors spend the majority of this section proposing a new mechanism to explain the seasonality of the fluoride signal. I do not think that the authors need to explain the origin of the fluoride signal to defend its use in their dating methodology, as the presence of its seasonal cycle is plainly evident regardless of its cause. That said, it is suggested that the fluoride seasonality is linked to sea ice seasonality and the behavior of East Antarctic polynyas. While I do find their hypothesis compelling, the authors do not provide sufficient observational or modeling results to support this hypothesis. I believe that it warrants a deeper investigation to be included in this study, especially given that the authors point out the many uncertainties in their interpretation, including (1) the difficulty in measuring fluoride due to its low concentrations, (2) the inconsistency of its reliability at different periods of the MBS record, (3) the volatility of fluoride, and (4) the existence of alternative explanations for the seasonal signal. Perhaps, as the authors suggest in Lines 394-5, a separate study examining sea ice proxies in the MBS cores would be a better place to introduce and test this hypothesis.

These are all fair points, and are raised in a largely similar fashion by the other reviewer of this manuscript. Given this, we will comprehensively revise this section to remove the bulk of the larger discussion around the sources of fluoride at MBS. We will re-investigate the sources and seasonality of fluoride in a separate manuscript as suggested.

Besides this one issue in Section 4.1, I found that the paper is well-written. This paper will be very useful for future analyses of the MBS ice cores, which will be a valuable archive of East Antarctic climate proxy records. Additionally, given the authors' thorough discussion of their methods, this paper will be a useful community document detailing the chemical analysis of and chronology development of annually resolved ice cores more broadly.

I recommend its publication after resolving the above point and the following minor points:

- Section 1: I appreciated the extensive discussion of site characteristics, but I wondered why the authors spent so much space discussing the wind characteristics? I don't believe that they ever returned to this later on in the discussion of the chronology development.

This manuscript was envisaged as not just a discussion of how the MBS chronology was developed, but also to have a general discussion around dating error and uncertainty, and its root causes. MBS is a high wind site (at least in comparison to other east Antarctic records we are familiar with) and we suspect this high

wind regime has a lot to do with the episodic nature of the accumulation at MBS (e.g. Jackson et al., 2023), the seasonal cycle of accumulation (Crockart et al., 2021), and the fact that there is likely to be frequent erosion events that will remove detail from the record (limiting the ability to perceive annual layers). Thus, understanding mean wind speed and direction during high accumulation and low accumulation periods, at least at the seasonal level, is critical to understanding where we might put a foot wrong in the dating process, so to speak. We wish to keep the wind characteristics discussion in the general site description section as we think it is highly relevant to understanding the root causes of dating errors, and the wind rose data is highly relevant to understanding the site features (e.g. sastrugi, dunes) that are evident in Figure 1b, as we do return to the discussion of how the site is likely quite episodic later in the manuscript.

- Figure 1, caption: I suggest the authors replace "fuschia" with "red" or "pink" and "cyan" with "blue" as these would be more universal color labels.

We will do this

- Line 111: It is stated that there is a mean sample resolution of 10 samples y-1. As resolution decreases with depth, it might be more useful to give a range of sample resolutions along the core (i.e., perhaps a mean sample resolution for a section of core near the top and a mean sample resolution for a section of core near the bottom).

We will do this

- Section 2.4: It would be useful to include a schematic of the CFA melter system used, either in the Appendix or as a main figure. If space limitations are a concern, I would suggest that Figure 3 could be combined with a CFA schematic into one new, slightly larger figure. See example CFA schematics in Figure 2 of Hoffman et al. (2022) or Figure 2 of Osterberg et al. (2006), among other papers.

We agree that a schematic of the CFA system would be useful, however the CFA system used for the MBS analyses was a modified version of that used by the Danish team in the past. As a result, two papers specifically detailing the CFA setup and datasets for both MBS impurities (Harlan, Kjær et al., in prep) and MBS isotopes (Gkinis et al., in prep) are being prepared and are close to submission. The modifications to the CFA system, which was used to process the MBS record therefore deserves a detailed study, including schematics of the new CFA setup which will be included in the Harlan, Kjær et al manuscript. It would not be appropriate to include those schematics here, as they are currently unpublished, but we can make reference to these upcoming papers in the manuscript.

- Line 228: The proper name for the "National Ice Core Facility" is the "National Science Foundation Ice Core Facility" (previously, it was named the "National Ice Core Laboratory," hence the confusion).

We will correct this

- Section 2.8, and Line 480, and Line 505: A simple schematic or table, even in the Appendix, illustrating the authors' 4-step dating method would be useful. I reread the dating section a few times and still struggle to understand when various chemical species were used in different dating schemes.

Thank you, this is a good idea -  we will devise a schematic to assist the reader.

- Line 365: I believe the authors mean "DMS" (dimethyl sulfide) when they write "MSA". Algae produce DMS, not MSA, and DMS is oxidized in the atmosphere into MSA and/or non-sea-salt

sulfate, among other products and intermediates (see Figure 1 of Fung et al. (2022) for a nice overview of DMS oxidation chemistry).

Yes, We will correct this, thank you.

- Figure 5: It would be useful to include row labels ("Main core non-satellite era", "Main core satellite era", "Charlie surface core") at the side similar to the column labels at the top.

We will do this

- Figure 5: The y-axis limits on the left 3 columns are such that the seasonal cycle is very clear, but for fluoride the axis limits are much wider than the fluoride seasonal signal, making its seasonality stand out less. I am wondering why the authors chose to minimize the apparent magnitude of its seasonal signal?

The fluoride concentrations are very low to begin with, and it seemed disingenuous to not show the smaller stature of the seasonal cycle. All the y-axes start at zero for the other species, so it might be considered misleading to not have the same axes for the fluoride. This shows that while a seasonal cycle is present, the low concentrations mean that it may not be analytically detectable from one year to the next. We would prefer to leave the fig 5 axes as is.

- Lines 455-8: The authors suggest that a key difference between the MBS and Law Dome ice cores is that MBS exhibits clear seasonality of fluoride, but then note that there is no fluoride dataset from Law Dome. I may be misunderstanding this, but the authors should not call this a difference between the sites if there is no evidence for the lack of fluoride seasonality at Law Dome. If they mean that this is not a difference between the site characteristics, but only a difference in how ice cores from the two sites were dated, this should be specified.

That's fair and confusing as currently written – we will re-write this to be clearer. There is a very small dataset of fluoride from Law Dome, but it is not yet processed or worked up, so we cannot compare the pair properly yet.

- Line 472: couldn't the hypothesis that there was a "training" period for the researchers near the top of the core be eliminated by repeating the layer-counting, at least for the top section of core?

Yes, however the point of the comparison to the layer counted only effort was to give a kind of 'worst case scenario' for missed layers. We will re-write this section to be clearer that that is what we meant.

- Figure B1: Y-axis labels for each series would be very useful.

The units are micro Equivalents per liter. We will add this to the caption.

**Typographical errors:**

We will fix these typos (below)

- Lines 90: I believe the word "was" is missing between "weight" and "recorded"
- Line 132: "non sea salt component" should be changed to "non-sea-salt component" in order to maintain consistency with the rest of the usages of "non-sea-salt" throughout the paper
- Line 179: there is a closing parenthesis, ")", where there should be none after the word "capillary"
- Table 3, caption: there is a missing closing parenthesis, ")" after "(see Plummer et al. (2012)"

- Lines 349-51: The usage of "e.g."'s are not consistent. In Line 349, e.g. begins a parenthetical, whereas in line 350, e.g. comes after a comma. I believe they both should begin parentheticals

- Line 361: "MBS-main" should have a capitalized "M" in "main"

- Line 442: There is an extra opening parenthesis, "(", before "van Ommen"

**References**

Fung, Ka Ming, Colette L. Heald, Jesse H. Kroll, Siyuan Wang, Duseong S. Jo, Andrew Gettelman, Zheng Lu, et al. "Exploring Dimethyl Sulfide (DMS) Oxidation and Implications for Global Aerosol Radiative Forcing." *Atmospheric Chemistry and Physics* 22, no. 2 (February 1, 2022): 1549–73. https://doi.org/10.5194/acp-22-1549-2022.

Hoffmann, Helene M., Mackenzie M. Grieman, Amy C. F. King, Jenna A. Epifanio, Kaden Martin, Diana Vladimirova, Helena V. Pryer, et al. "The ST22 Chronology for the Skytrain Ice Rise Ice Core – Part 1: A Stratigraphic Chronology of the Last 2000 Years." *Climate of the Past* 18, no. 8 (August 10, 2022): 1831–47. https://doi.org/10.5194/cp-18-1831-2022.

Osterberg, Erich C., Michael J. Handley, Sharon B. Sneed, Paul A. Mayewski, and Karl J. Kreutz. "Continuous Ice Core Melter System with Discrete Sampling for Major Ion, Trace Element, and Stable Isotope Analyses." *Environmental Science & Technology* 40, no. 10 (May 1, 2006): 3355–61. https://doi.org/10.1021/es052536w.

---

## Author Comment (AC2)

Referee 2 Holly Winton

Vance et al. present the chronology for a new ~300 m ice core recovered from Mount Brown South in East Antarctica filling a void in the spatial array of ice core records in the region. The manuscript describes the drilling, processing and analytical procedures and presents the age-depth model. The core was dated using a range of annually resolved chemical species and a number of volcanic tie points. I enjoyed reading about the group approach to annual layer counting and the use of two independent annual layer counts to derive the final age-depth model. The authors also describe the seasonality of fluoride in the core. Fluoride is rarely detected in Antarctic ice cores and in the atmosphere over the Southern Ocean and thus the sources and atmospheric processes of fluoride in the region are not well understood. Below are suggestions that I hope will improve the manuscript before publication in Climate of the Past.

**Main comments**

Ice chemistry analysis and figures of merit

The ICP-MS methods section reports figures of merit including LOD and reproducibility. Some LOD data are presented in Table 4. Please report accuracy of the ICP-MS measurements. Did you measure sulphur or aluminium via ICP-MS? As you have done for the ICP-MS, please report figures of merit for IC and CFA measurements including blank concentrations, accuracy, precision and also note the concentration range of calibration standards. How do sodium and calcium concentrations compare between ICP-MS, IC and CFA measurements?

There are multiple aspects to this comment, which we have separated out to address individually below (in italics). Because this paper uses data from already published methods, we opted to refer the reader to the publications that detail these methods, including the full figures of merit. We don't think it is within the scope of this work to reproduce the full figures of merit (such as calibration ranges and blank concentrations). However, we acknowledge the captions for Table 4, and the discussion around concentrations didn't clearly identify where these figures of merit from the method development papers could be found. We will re-write these sections to make this clearer and include figures of merit for the IC data where it makes sense.

*Please report accuracy of the ICP-MS measurements.*

These are reported in Vallelonga et al., 2017 and associated previous publications for the method. Note that the ICP-MS data was not used to develop the chronology reported here, so we don't think it appropriate to report in detail in this work (the ICP-MS data is being prepared for publication elsewhere). We will revise the caption to ensure the reader is clearly directed to the appropriate publications for details around the analytical method and associated accuracy.

*Did you measure sulphur or aluminium via ICP-MS?*

No we didn't.

*As you have done for the ICP-MS, please report figures of merit for IC and CFA measurements including blank concentrations, accuracy, precision and also note the concentration range of calibration standards.*

We don't report CFA impurities measurements in this manuscript, as the CFA data is being prepared for publication elsewhere, and the data was not ready for use. We used the discretely derived chemistry and stable water isotope ratios exclusively for dating and development of the chronologies here (with the bulk of emphasis on the chemistry, as it contained the clearest annual layers). We also looked at an interim CFA water isotope ratios dataset for confirmatory purposes during dating, but this was not on a finalised depth scale, so it wasn't specifically investigated in the matchmaker dataset. The figures of merit for the IC data

will be reported in this new CFA data paper, which will be a comprehensive analysis of the current (modified) UCPH / Niels Bohr CFA system used. We will ensure the reader is clearly directed to the figures of merit for IC data.

*How do sodium and calcium concentrations compare between ICP-MS, IC and CFA measurements?*

We suggest this would be better explored in the MBS ICP-MS paper that is being developed by the co-authors here that have developed that dataset, as there is likely to be some measure of discrepancies between the three techniques (ICP-MS\IC\CFA sodium and calcium). This is because IC and CFA determine only the soluble fraction, while ICP-MS can determine also a fraction of the insoluble\mineral component for these elements. The reasons behind these discrepancies deserve a full analysis, including any difference between soluble and insoluble fractions. The point of showing the mean concentrations here is to have a brief discussion around any similarities and differences between the two sites (Law Dome and MBS) as the datasets from both sites have been measured using the same methods. Thus, they are directly comparable across sites, but not necessarily across analytical techniques.

As above the CFA impurities data is under development. The MBS chronology was primarily developed using the discrete chemistry data measured in Hobart. It should also be noted that the different analyses have been measured at very different resolutions (~25 cm for ICP-MS, compared to 3cm for discrete chemistry).

Depth scale

As the focus of this manuscript is the age-depth scale of the MBS core, I encourage the authors to include the description of the scaling and shift factors. This would be useful for the community as discrepancies between field and lab depth scales and core breakage is not unique to the MBS core. Can you estimate a depth uncertainty of the master depth model?

Scaling and shift factors are being actively employed in the development of the CFA impurities and CFA isotopes datasets, which will both be published as data descriptor papers. For this study we can certainly include a description of how the Hobart depth model was aligned to the UCPH CFA stick lengths in the revised document. We propose this to be in the form of a simple shift/scale correction (equation) that will describe how a discrepancy in (Hobart) core lengths and (UCPH) CFA stick lengths was solved for each core / bag.

By depth uncertainty, we assume the reviewer means the differences between field (drilling) depths and bag lengths recorded during discrete sample processing? This is detailed in section 2.7, lines 248-250:

*To solve the length discrepancies and derive a master depth model, we compared field and lab measurements to the ILCS scans to derive the correct length of each core. This painstaking process was then used to determine the full drilled depth of 294.785 metres, which is 18.5 cm different from the field measured depth of 294.6 metres (i.e. a 0.003% error).*

Age uncertainty

An age uncertainty of ±2 years is reported in the conclusions. How was this derived? Please report in the abstract and main text.

The age uncertainty is derived from the WAIS Divide uncertainty, given we have synchronised MBS to WAIS (Table 3 and section 3.1) We will report the synchronization in the abstract as well.

Fluoride

The detection of fluoride and its seasonality is an interesting finding. Given fluoride has a different seasonality to the other markers, it is helpful to identify the annual layers and thus useful in this context. Yet, fluoride is largely unexplored in Antarctic ice cores and the modern atmosphere over the Southern Ocean so much so that we know little about the sources and photochemical processes in this unique and pristine environment and without this understanding, interpreting ice core fluoride is largely speculative. A study understanding the air-snow transfer of fluoride and the post-depositional processes along with exploring ice core fluoride relationships with a range of climate variables over the instrumental era would be incredibly valuable to further understand the potential as a sea ice proxy. Given fluoride is volatile, the first step is to understand how photochemistry between the atmosphere and surface snow impacts the archived fluoride signal. For example, over a decade has been dedicated to understanding these processes for ice core nitrate. Snice this information is currently lacking for ice core fluoride, I suggest focussing manuscript on the use of fluoride as an annual marker and moving the discussion on fluoride as a potential sea ice proxy to a separate manuscript dedicated to understanding fluoride deposition at the MBS south.

Noted, as with the other reviewer, we will comprehensively shorten and revise this fluoride section to remove speculation around the sources of fluoride.

The detection of fluoride in MBS raises many questions. For example, what were the summer and winter concentrations of fluoride in the MSB core and how do they compare to Severi et al. (2014) and Morganti et al. (2007). Is the seasonality the same between the three studies? How does the seasonality of fluoride compare to nitrate, bromine and iodine which also undergo photochemical/ post-depositional processes? What information is known about fluoride from Southern Ocean aerosol studies? I understand, from personal communication, aerosol fluoride also exhibits a seasonal cycle at the Cape Grim Baseline Air Pollution Station.

These are excellent questions for a manuscript focussed on the fluoride data. As suggested we will greatly reduce this section in a revised manuscript and focus on the dating here. We thank the reviewer for the questions they pose that could be explored. I was not aware of the fluoride sampling at Cape Grim!

Data availability

Note that the age-depth model not supplied and not yet available on the Australian Antarctic Data Centre.

This data will be uploaded to the Australian Antarctic Data Centre prior to the submission of the revised manuscript, such that a doi / link is available for the data if the manuscript is accepted.

**Specific comments**

L78 water isotopic ratios (here and throughout) We will correct this

L105 and L120 how are the 3 cm resolution chemistry samples mentioned here different to the 3.5 cm chemistry samples mentioned in L107? We will re-write this – we acknowledge it is quite confusing as it is currently written. The 3.5 cm sample refers to the analysed section of the final 4cm sample in a 1 metre bag (e.g 1 metre if ice yields 32 x 3 cm samples (96cm) plus 1 x 4cm sample, which becomes 3.5 cm after sampling in our laminar flow vice system.

L105-117 how many samples per year result from this sampling resolution? This is written at line 111 – 10 samples per year. As with reviewer one, we will provide a range for this annual sample resolution.

L124-125 how did you mitigate this? This organic contamination could be drill fluid contamination which has been observed in some ice core samples where drill fluid has contaminated the core through micro fractures and impacts the shoulder of the MSA peak. We think this was laboratory contamination. It is

unlikely to have been drill fluid, as this contamination occurred in the dry drilled section of the core (~20-93 metres). We will modify this sentence to be clear where we think the contamination came from.

L154 hydrogen peroxide We will correct this

L167 sodium We will correct this

L170 number of particles. Add reference. We will correct this

L172 add resistivity of Milli-Q water. We will correct this

L173 calibration standards Noted, we will revise accordingly

L177 manuscript uses both "mL" and "ml" We will correct this

L180 CRDS We will define this acronym

L185 delete 'halogens'? We will correct this

L185 how were these sub-sampled? Noted, we will revise accordingly.

L195 ICP-MS tubing? Noted, we will revise accordingly

L205-206 reported in Table 4 Noted, we will revise accordingly

L268-270 references required here to justify assignment of these peaks to 1 January. We think this is explained in the following sentences about the difference between Law Dome and MBS annual horizon dates. We will revisit this section to ensure clarity.

L313 in the case of an uncertain counted year, where did you place the annual marker? e.g. on the nss-sulphate or water isotope peak? This depended on the available evidence in each case. We will revise to make this clear.

L330 e.g., extreme precipitation events (Turner et al. 2019) Noted, we will revise accordingly

L423 which is the "prior study"? Foster et al., 2006 (already cited here and in introduction).

L429 MSA is a proxy of sea ice in some regions of Antarctica. Noted, we will revise accordingly

Figure 1 A scale bar on panel b would be helpful to see the extent of the snow features. Add snow pit label to panel b. We will correct this

Figure 3 Add dimensions We will correct this

Figure 4 Y-axis label missing We will correct this

---

## Author Response (AR1)

Author responses for Referee1 – Jacob Chalif

In the paper "An annually resolved chronology of the Mount Brown South ice cores, East Antarctica," T.R. Vance and coauthors present four new chronologies for ice cores (three surface cores and one deeper core) from Mount Brown South (MBS) in East Antarctica. These chronologies were developed through a combined multi-researcher annual chemical layer counting and volcanic alignment approach, which is common in dating annually resolved ice core records. Preliminary analyses of ice core chemistry, in particular comparisons with Law Dome and an analysis of the seasonal cycle of certain chemical species used in the annual layer counting effort, are presented as well.

I applaud the authors for their comprehensive discussion of the methods used in analyzing the MBS ice cores. Not all ice core studies are so transparent in detailing their methods of analysis, but the authors do a commendable job of laying out their complete analytical regime, which involved multiple cores, institutions, and types of chemical and physical analyses.

Given the quality of the annual layers in the MBS ice cores, I completely agree with the method the authors chose to use to establish the MBS chronologies.

In Section 4.1, the seasonal cycles of trace chemical species are discussed. This discussion is foundationally important for their dating methodology, but the authors spend the majority of this section proposing a new mechanism to explain the seasonality of the fluoride signal. I do not think that the authors need to explain the origin of the fluoride signal to defend its use in their dating methodology, as the presence of its seasonal cycle is plainly evident regardless of its cause. That said, it is suggested that the fluoride seasonality is linked to sea ice seasonality and the behavior of East Antarctic polynyas. While I do find their hypothesis compelling, the authors do not provide sufficient observational or modeling results to support this hypothesis. I believe that it warrants a deeper investigation to be included in this study, especially given that the authors point out the many uncertainties in their interpretation, including (1) the difficulty in measuring fluoride due to its low concentrations, (2) the inconsistency of its reliability at different periods of the MBS record, (3) the volatility of fluoride, and (4) the existence of alternative explanations for the seasonal signal. Perhaps, as the authors suggest in Lines 394-5, a separate study examining sea ice proxies in the MBS cores would be a better place to introduce and test this hypothesis.

These are all reasonable points, and are raised in a largely similar fashion by the other reviewer of this manuscript. Given this, we have comprehensively revised this section to remove the bulk of the larger discussion around the potential sources of fluoride at MBS – although we retain a few points since the detection of a seasonal fluoride signal is, as suggested here, of great interest to the community and it would be strange not to discuss it in at least a preliminary fashion. We will re-investigate the sources and seasonality of fluoride in a separate dedicated project as suggested. See revised section 4.1, lines 378-396 and lines 397-401

Besides this one issue in Section 4.1, I found that the paper is well-written. This paper will be very useful for future analyses of the MBS ice cores, which will be a valuable archive of East Antarctic climate proxy records. Additionally, given the authors' thorough discussion of their methods, this paper will be a useful community document detailing the chemical analysis of and chronology development of annually resolved ice cores more broadly.

I recommend its publication after resolving the above point and the following minor points:

- Section 1: I appreciated the extensive discussion of site characteristics, but I wondered why the authors spent so much space discussing the wind characteristics? I don't believe that they ever returned to this later on in the discussion of the chronology development.

This manuscript was envisaged as not just a discussion of how the MBS chronology was developed, but also to have a general discussion around dating error and uncertainty, and its root causes. MBS is a high wind site (at least in comparison to other East Antarctic records we are familiar with) and we suspect this high wind regime has a lot to do with the episodic nature of the accumulation at MBS (e.g. Jackson et al., 2023), the seasonal cycle of accumulation (Crockart et al., 2021), and the fact that there is likely to be frequent erosion events that will remove detail from the record (limiting the ability to perceive annual layers). Thus, understanding mean wind speed and direction during high accumulation and low accumulation periods, at least at the seasonal level, is critical to understanding where we might put a foot wrong in the dating process, so to speak. We wish to keep the wind characteristics discussion in the general site description section as we think it is highly relevant to understanding the root causes of dating errors, and the wind rose data is relevant to understanding the site features (e.g. sastrugi, dunes) that are evident in Figure 1b, as we do return to the discussion of how the site is likely quite episodic later in the manuscript. Resolved to leave as is.

- Figure 1, caption: I suggest the authors replace "fuschia" with "red" or "pink" and "cyan" with "blue" as these would be more universal color labels.

Done, See revised Figure 1 Caption

- Line 111: It is stated that there is a mean sample resolution of 10 samples y-1. As resolution decreases with depth, it might be more useful to give a range of sample resolutions along the core (i.e., perhaps a mean sample resolution for a section of core near the top and a mean sample resolution for a section of core near the bottom).

We have now provided a mean in the upper firn layers, as well as in the ice. Lines 114-115

- Section 2.4: It would be useful to include a schematic of the CFA melter system used, either in the Appendix or as a main figure. If space limitations are a concern, I would suggest that Figure 3 could be combined with a CFA schematic into one new, slightly larger figure. See example CFA schematics in Figure 2 of Hoffman et al. (2022) or Figure 2 of Osterberg et al. (2006), among other papers.

We agree that a schematic of the CFA system would be useful, however the CFA system used for the MBS analyses was a modified version of that used by the Danish team in the past. As a result, two papers specifically detailing the CFA setup and datasets for both MBS impurities (Harlan, Kjær et al., in prep) and MBS isotopes (Gkinis et al., in prep) are being prepared and are close to submission. The modifications to the CFA system, which was used to process the MBS record therefore deserves a detailed study, including schematics of the new CFA setup which will be included in the Harlan, Kjær et al manuscript. Note that we did not use CFA chemistry analyses to produce the chronologies detailed in this manuscript (only the discrete analyses form Hobart and Canberra, apart from the CFA isotopic ratios for the wet drilled core, which was used for comparison / confirmatory purposes only. We now state clearly in paragraph 3 of section 2.4 that the CFA set up and datasets are in preparation for publication. Lines 180-181, and line 191.

- Line 228: The proper name for the "National Ice Core Facility" is the "National Science Foundation Ice Core Facility" (previously, it was named the "National Ice Core Laboratory," hence the confusion).

Corrected, Line 235

- Section 2.8, and Line 480, and Line 505: A simple schematic or table, even in the Appendix, illustrating the authors' 4-step dating method would be useful. I reread the dating section a few

times and still struggle to understand when various chemical species were used in different dating schemes.

Thank you, this is a good idea - we have added a summary of the dating process in the appendix and now refer the reader to this at line 302 and see Appendix C.

- Line 365: I believe the authors mean "DMS" (dimethyl sulfide) when they write "MSA". Algae produce DMS, not MSA, and DMS is oxidized in the atmosphere into MSA and/or non-sea-salt sulfate, among other products and intermediates (see Figure 1 of Fung et al. (2022) for a nice overview of DMS oxidation chemistry).

Yes, this was a typo, we did mean DMS. Corrected line 380

- Figure 5: It would be useful to include row labels ("Main core non-satellite era", "Main core satellite era", "Charlie surface core") at the side similar to the column labels at the top.

We tried this, but it made the figure very 'busy', as the time periods differ depending on the species. Instead, we have stated in the opening sentences of the caption now exactly which time period and core is being referred to by each row. Hopefully this is clearer. See caption for figure 5, first 3 sentences.

- Figure 5: The y-axis limits on the left 3 columns are such that the seasonal cycle is very clear, but for fluoride the axis limits are much wider than the fluoride seasonal signal, making its seasonality stand out less. I am wondering why the authors chose to minimize the apparent magnitude of its seasonal signal?

The fluoride concentrations are very low to begin with, and it seemed disingenuous to not show the smaller stature of the seasonal cycle. All the y-axes start at zero for the other species, so it might be considered misleading to not have the same axes commence at zero for the fluoride. This shows that while a seasonal cycle is present, the low concentrations mean that it may not be analytically detectable from one year to the next. For this reason, we would prefer to leave the axes as is.

- Lines 455-8: The authors suggest that a key difference between the MBS and Law Dome ice cores is that MBS exhibits clear seasonality of fluoride, but then note that there is no fluoride dataset from Law Dome. I may be misunderstanding this, but the authors should not call this a difference between the sites if there is no evidence for the lack of fluoride seasonality at Law Dome. If they mean that this is not a difference between the site characteristics, but only a difference in how ice cores from the two sites were dated, this should be specified.

We agree – that was quite confusing as originally written! We do have a very small dataset of fluoride from Law Dome, but it is not yet definitively processed and quality controlled, so we cannot compare the pair properly yet. We have re-written this sentence to say:

 "A final comparison between MBS and Law Dome will be to establish whether fluoride concentrations at Law Dome contain a seasonal cycle, as they do at MBS, however this will rely on the development of a robust fluoride dataset from Law Dome to substantiate differences between the two sites." Lines 456-459

- Line 472: couldn't the hypothesis that there was a "training" period for the researchers near the top of the core be eliminated by repeating the layer-counting, at least for the top section of core?

Perhaps, however the point of the comparison to the layer counted only effort was to give a kind of 'worst case scenario' for missed layers. In addition, we already suggest two further reasons that we think contribute to this greater divergence, one of which is the greater number of samples per year may have led

to noisier data, while the second is missing fluoride data in the upper 200 years, which given F was pivotal to the layer counting in MBSHK, would not be solved by a recount. We have re-phrased the final paragraph of this section, to hopefully clarify this. Please see section 4.3, lines 474-480 and also lines 485-488

- Figure B1: Y-axis labels for each series would be very useful.

The units are micro Equivalents per litre. This is now added to the caption. To add two y axes would make it difficult to make out detail in the matchmaker generated image (which would have to be correspondingly smaller), so we have opted to state this in the caption rather than as axes. See first sentence in the caption for figure B1.

**Typographical errors:**

We will fix these typos (below)

- Lines 90: I believe the word "was" is missing between "weight" and "recorded" Corrected, line 93

- Line 132: "non sea salt component" should be changed to "non-sea-salt component" in order to maintain consistency with the rest of the usages of "non-sea-salt" throughout the paper Corrected, line 136

- Line 179: there is a closing parenthesis, ")", where there should be none after the word "capillary" Corrected, line 186

- Table 3, caption: there is a missing closing parenthesis, ")" after "(see Plummer et al. (2012)" Corrected, Table 3 caption

- Lines 349-51: The usage of "e.g."'s are not consistent. In Line 349, e.g. begins a parenthetical, whereas in line 350, e.g. comes after a comma. I believe they both should begin parentheticals Removed unnecessary e.g.'s here and elsewhere line 363

- Line 361: "MBS-main" should have a capitalized "M" in "main" Corrected, line 376

- Line 442: There is an extra opening parenthesis, "(", before "van Ommen" Corrected, line 443

**References**

Fung, Ka Ming, Colette L. Heald, Jesse H. Kroll, Siyuan Wang, Duseong S. Jo, Andrew Gettelman, Zheng Lu, et al. "Exploring Dimethyl Sulfide (DMS) Oxidation and Implications for Global Aerosol Radiative Forcing." *Atmospheric Chemistry and Physics* 22, no. 2 (February 1, 2022): 1549–73. https://doi.org/10.5194/acp-22-1549-2022.

Hoffmann, Helene M., Mackenzie M. Grieman, Amy C. F. King, Jenna A. Epifanio, Kaden Martin, Diana Vladimirova, Helena V. Pryer, et al. "The ST22 Chronology for the Skytrain Ice Rise Ice Core – Part 1: A Stratigraphic Chronology of the Last 2000 Years." *Climate of the Past* 18, no. 8 (August 10, 2022): 1831–47. https://doi.org/10.5194/cp-18-1831-2022.

Osterberg, Erich C., Michael J. Handley, Sharon B. Sneed, Paul A. Mayewski, and Karl J. Kreutz. "Continuous Ice Core Melter System with Discrete Sampling for Major Ion, Trace Element, and Stable Isotope Analyses." *Environmental Science & Technology* 40, no. 10 (May 1, 2006): 3355–61. https://doi.org/10.1021/es052536w.

Author responses to Referee 2 – Holly Winton

Vance et al. present the chronology for a new ~300 m ice core recovered from Mount Brown South in East Antarctica filling a void in the spatial array of ice core records in the region. The manuscript describes the

drilling, processing and analytical procedures and presents the age-depth model. The core was dated using a range of annually resolved chemical species and a number of volcanic tie points. I enjoyed reading about the group approach to annual layer counting and the use of two independent annual layer counts to derive the final age-depth model. The authors also describe the seasonality of fluoride in the core. Fluoride is rarely detected in Antarctic ice cores and in the atmosphere over the Southern Ocean and thus the sources and atmospheric processes of fluoride in the region are not well understood. Below are suggestions that I hope will improve the manuscript before publication in Climate of the Past.

**Main comments**

Ice chemistry analysis and figures of merit

The ICP-MS methods section reports figures of merit including LOD and reproducibility. Some LOD data are presented in Table 4. Please report accuracy of the ICP-MS measurements. Did you measure sulphur or aluminium via ICP-MS? As you have done for the ICP-MS, please report figures of merit for IC and CFA measurements including blank concentrations, accuracy, precision and also note the concentration range of calibration standards. How do sodium and calcium concentrations compare between ICP-MS, IC and CFA measurements?

There are multiple aspects to this comment, which we have separated out to address individually below (in italics). Please see sections below for specific revisions.

*Please report accuracy of the ICP-MS measurements.*

These are already reported in Vallelonga et al., 2017 and associated previous publications for the method. We have reproduced some of these values – e.g. blank values, detection limits and accuracy / precision, in the ICP-MS methods at lines 205-213. Note that the ICP-MS data was not used to develop the chronology reported here, so we don't think it appropriate to report in further detail in this work (the ICP-MS data is being prepared for publication elsewhere).

For the IC data, we have revised the caption to ensure the reader is clearly directed to the appropriate publications for details around the analytical methods and associated accuracy and other method validations. We have also created a new table in the appendix which specifically details  standard concentrations ranges, mean blank values and detection limits for each species analysed via IC. See Table 4 caption and appendix table D1

*Did you measure sulphur or aluminium via ICP-MS?*

No we didn't – we only measured Br, I, Na and Ca.

*As you have done for the ICP-MS, please report figures of merit for IC and CFA measurements including blank concentrations, accuracy, precision and also note the concentration range of calibration standards.*

We don't report CFA impurities measurements in this manuscript, as the CFA data is being prepared for publication elsewhere. We only used the discretely derived chemistry (IC) and stable water isotope ratios for dating and development of the chronologies here (with the bulk of emphasis on the chemistry, as it contained the clearest annual layers). For the wet drilled section (CFA isotopes only) we viewed an interim CFA water isotope ratios dataset for confirmatory purposes of our layer counting during dating. Thus, the figures of merit for the CFA data will be reported in the CFA data paper, which will be a comprehensive analysis of the current (modified) UCPH / Niels Bohr CFA system used.

For the IC data, we have revised the caption of Figure 4 as above to ensure it is clear where these figures of merit have been previously published, and we have now added a table in the appendix to specifically provide figures of merit for IC data, as with our reply above. See appendix table D1

*How do sodium and calcium concentrations compare between ICP-MS, IC and CFA measurements?*

We suggest this would be better explored in the MBS ICP-MS paper that is being developed by the co-authors here that have developed that dataset, as there is likely to be some measure of discrepancies between the three techniques (ICP-MS\IC\CFA sodium and calcium). This is because IC and CFA determine only the soluble fraction, while ICP-MS can determine also a fraction of the insoluble\mineral component for these elements. In addition, the ICP-MS analyses are 25 cm mean resolution, compared to the 3 cm for the IC data investigated in this study. The reasons behind these discrepancies deserve a full analysis, including any difference between soluble and insoluble fractions. The point of showing the mean concentrations here is to have a brief discussion around any similarities and differences between the two sites (Law Dome and MBS) as the datasets from both sites have been measured using the same methods. Thus, they are directly comparable across sites, but not necessarily across analytical techniques. Nonetheless, we have now provided mean values for the ICP-MS measurements of calcium and sodium, which are reasonably comparable to the IC values for both MBS and Law Dome despite the caveats above. See Table 4.

Depth scale

As the focus of this manuscript is the age-depth scale of the MBS core, I encourage the authors to include the description of the scaling and shift factors. This would be useful for the community as discrepancies between field and lab depth scales and core breakage is not unique to the MBS core. Can you estimate a depth uncertainty of the master depth model?

Scaling and shift factors are being actively employed in the development of the CFA impurities and CFA isotopes datasets, which will both be published as data descriptor papers. For this study using the discrete 3 cm data, we have included a description of how we averaged the difference between laboratory and final accepted core lengths across the sample depths within the core, which ensures the uncertainty is distributed across all samples rather than accumulating as a final error in the final sample of each core. See lines 254-256

By depth uncertainty, we assume the reviewer means the differences between field (drilling) depths and bag lengths recorded during discrete sample processing? This is detailed in section 2.7, lines 248-250:

*To solve the length discrepancies and derive a master depth model, we compared field and lab measurements to the ILCS scans to derive the correct length of each core. This painstaking process was then used to determine the full drilled depth of 294.785 metres, which is 18.5 cm different from the field measured depth of 294.6 metres (i.e. a 0.003% error).*

Age uncertainty

An age uncertainty of ±2 years is reported in the conclusions. How was this derived? Please report in the abstract and main text.

The age uncertainty is derived from the WAIS Divide uncertainty, given we have synchronised MBS to WAIS (WD2014) For further information, see caption for Table 3 and section 3.1, Lines 329-334. This is also reported in the main text at the end of section 2.8, lines 300-302

Fluoride

The detection of fluoride and its seasonality is an interesting finding. Given fluoride has a different seasonality to the other markers, it is helpful to identify the annual layers and thus useful in this context. Yet, fluoride is largely unexplored in Antarctic ice cores and the modern atmosphere over the Southern Ocean so much so that we know little about the sources and photochemical processes in this unique and pristine environment and without this understanding, interpreting ice core fluoride is largely speculative. A study understanding the air-snow transfer of fluoride and the post-depositional processes along with exploring ice core fluoride relationships with a range of climate variables over the instrumental era would be incredibly valuable to further understand the potential as a sea ice proxy. Given fluoride is volatile, the first step is to understand how photochemistry between the atmosphere and surface snow impacts the archived fluoride signal. For example, over a decade has been dedicated to understanding these processes for ice core nitrate. Snice this information is currently lacking for ice core fluoride, I suggest focussing manuscript on the use of fluoride as an annual marker and moving the discussion on fluoride as a potential sea ice proxy to a separate manuscript dedicated to understanding fluoride deposition at the MBS south.

Noted, as with the other reviewer, we have comprehensively revised this section to remove the bulk of the larger discussion around the potential sources of fluoride at MBS – although we retain a few points since the detection of a seasonal fluoride signal is, as suggested here, of great interest to the community and it would be strange not to discuss it in at least a preliminary fashion. We will re-investigate the sources and seasonality of fluoride in a separate manuscript as suggested. For now, we make initial mention of the distinct seasonal cycle (e.g. compared to sea salts) before acknowledging the many pre- and post-depositional processes as well as analytical constraints that would need to be accounted for in the interpretation of any climate signal from fluoride. See revised section 4.1, lines 378-396 and lines 397-401

The detection of fluoride in MBS raises many questions. For example, what were the summer and winter concentrations of fluoride in the MSB core and how do they compare to Severi et al. (2014) and Morganti et al. (2007). Is the seasonality the same between the three studies? How does the seasonality of fluoride compare to nitrate, bromine and iodine which also undergo photochemical/ post-depositional processes? What information is known about fluoride from Southern Ocean aerosol studies? I understand, from personal communication, aerosol fluoride also exhibits a seasonal cycle at the Cape Grim Baseline Air Pollution Station.

These are excellent questions for a manuscript focussed on the fluoride data. As suggested we will greatly reduce this section in a revised manuscript and focus on the dating here. We thank the reviewer for the questions they pose that could be explored, and the information of the fluoride sampling at Cape Grim.

Data availability

Note that the age-depth model not supplied and not yet available on the Australian Antarctic Data Centre.

This dataset and the data used to produce the chronology will be uploaded to the Australian Antarctic Data Centre and will be made publicly available upon acceptance of the manuscript. The doi is now provided in the data availability section. Lines 520-522

**Specific comments**

L78 water isotopic ratios (here and throughout) Corrected at lines 7, 44, 81, 146, 149, 445, 470 Table 2 caption.

L105 and L120 how are the 3 cm resolution chemistry samples mentioned here different to the 3.5 cm chemistry samples mentioned in L107?

The 3.5 cm sample refers to the analysed section of the final 4cm sample in a 1 metre bag (e.g  1 metre of ice core yields 32  3 cm samples (96cm) plus 1 x 4cm sample, which becomes a 3.5 cm chemistry sample after cleaning in our laminar flow vice system. This is explained at lines 108-113

L105-117 how many samples per year result from this sampling resolution? This is written at line 114 – it is 10 samples per year. As with the request from reviewer 1, we have provided a range for this annual sample resolution in firn and ice. Please see lines 114-115

L124-125 how did you mitigate this? This organic contamination could be drill fluid contamination which has been observed in some ice core samples where drill fluid has contaminated the core through micro fractures and impacts the shoulder of the MSA peak.

We think this was laboratory contamination. It can't have been drill fluid contamination, as it occurred in the dry drilled section of the core prior to wet drilling, and the cores have always been kept separated. (~20-93 metres). We mitigated by cutting an entirely new chemistry stick for subsequent analysis as detailed in lines 127-130

L154 hydrogen peroxide Corrected, Line 173

L167 sodium Corrected, Line 172

L170 number of particles. Add reference. Reference will be the in preparation CFA paper referred to at the end of the paragraph.

L172 add resistivity of Milli-Q water. Corrected, Line 177

L173 calibration standards ? This already appears as is.

L177 manuscript uses both "mL" and "ml" Corrected to mL, Line 184

L180 CRDS Corrected, Line 187

L185 delete 'halogens'? Corrected, Line 198

L185 how were these sub-sampled? Corrected to include 'via a dedicated CFA line to a fraction collector' Corrected, Lines…

L195 ICP-MS tubing? Corrected, Line 209

L205-206 reported in Table 4 These results are referred to by table in the results section. We think that is more appropriate than referring to the results here in the methods. We have left as is in this case.

L268-270 references required here to justify assignment of these peaks to 1 January. We think this thinking is explained in the following sentences about the difference between Law Dome and MBS annual horizon dates. However, we have added a reference to Crockart et al., which defined layer counting of the satellite era section of MBS ice cores (in conjunction with comparison to reanalyses) with a horizon date of 1 January. See revised section, Line 287

L313 in the case of an uncertain counted year, where did you place the annual marker? e.g. on the nss-sulphate or water isotope peak? This depended on the available evidence in each case. We have revised to make this clear by adding 'In the former case of uncertain counted years, the annual horizon was placed where the dating team deemed the most evidence for its placement to be.' Lines 321-322

L330 e.g., extreme precipitation events (Turner et al. 2019) Added reference, Line 340

L423 which is the "prior study"? Foster et al., 2006 (already cited here and in introduction). Corrected to 'the prior study' Line 452

L429 MSA is a proxy of sea ice in some regions of Antarctica. Corrected, Line 460

Figure 1 A scale bar on panel b would be helpful to see the extent of the snow features. Add snow pit label to panel b. Have added in the caption that a snowpit was dug in the same location as Bravo and Charlie cores. See revised caption for Figure 1. A scale bar would not work on this image, as it was photographed from a twin otter at low tangential elevation, thus the scale changes throughout the image.

Figure 3 Add dimensions Added nominal dimensions to caption. See revised caption for Figure 3. Note that remaining dimensions of archive sections, as well as the planed ILCS slabs varied considerably depending on the quality of the remaining section itself and prior processing steps.

Figure 4 Y-axis label missing This is already indicated in the caption. See caption for Figure 4.